# Homotopy-based training of NeuralODEs for accurate dynamics discovery

**Joon-Hyuk Ko**[†]
Department of Physics & Astronomy
Seoul National University
Seoul, 08826, South Korea
jhko725@snu.ac.kr

**Hankyul Koh**[†]
Department of Physics & Astronomy
Seoul National University
Seoul, 08826, South Korea
physics113@snu.ac.kr

**Nojun Park**
Department of Physics
Massachusetts Institute of Technology
MA, 02142, United States
bnj11526@mit.edu

**Wonho Jhe**
Department of Physics & Astronomy
Seoul National University
Seoul, 08826, South Korea
whjhe@snu.ac.kr

## Abstract

Neural Ordinary Differential Equations (NeuralODEs) present an attractive way to extract dynamical laws from time series data, as they bridge neural networks with the differential equation-based modeling paradigm of the physical sciences. However, these models often display long training times and suboptimal results, especially for longer duration data. While a common strategy in the literature imposes strong constraints to the NeuralODE architecture to inherently promote stable model dynamics, such methods are ill-suited for dynamics discovery as the unknown governing equation is not guaranteed to satisfy the assumed constraints. In this paper, we develop a new training method for NeuralODEs, based on synchronization and homotopy optimization, that does not require changes to the model architecture. We show that synchronizing the model dynamics and the training data tames the originally irregular loss landscape, which homotopy optimization can then leverage to enhance training. Through benchmark experiments, we demonstrate our method achieves competitive or better training loss while often requiring less than half the number of training epochs compared to other model-agnostic techniques. Furthermore, models trained with our method display better extrapolation capabilities, highlighting the effectiveness of our method.

## 1   Introduction

Predicting the evolution of a time varying system and discovering mathematical models that govern it is paramount to both deeper scientific understanding and potential engineering applications. The centuries-old paradigm to tackle this problem was to either ingeniously deduce empirical rules from experimental data, or mathematically derive physical laws from first principles. However, the complexities of systems of interest have grown so much that these traditional approaches are now often insufficient. This has led to a growing interest in using machine learning methods to infer dynamical laws from data.

One line of research stems from the dynamical systems literature, where the problem of interest was to determine the unknown coefficients of a given differential equation from data. While the task seems rather simple, it was found that naive optimization cannot properly recover the desired coefficients

37th Conference on Neural Information Processing Systems (NeurIPS 2023).

due to irregularities in the optimization landscape, which worsens with increasing nonlinearity and data length [64, 50, 56, 1].

A parallel development occurred in the study of recurrent neural networks (RNNs), which are neural network analogues of discrete maps. It was quickly found that training these models can be quite unstable and results in exploding or vanishing gradients. While earlier works in the field resulted in techniques such as teacher forcing or gradient clipping that are agnostic to the RNN architecture [14, 45], more recent approaches have been directed towards circumventing the problem entirely by incorporating specific mathematical forms into the model architecture to constrain the eigenvalues of the model Jacobian and stabilize RNN dynamics [4, 29, 9, 34, 42]. While these latter methods can be effective, for the purpose of discovering dynamical equations underlying the data, these approaches are inadequate as one cannot ascertain the hidden equations conform to the predetermined rigid mathematical structures - especially so if the dynamics to be predicted is incompatible with the constrained model architecture [57, 41].

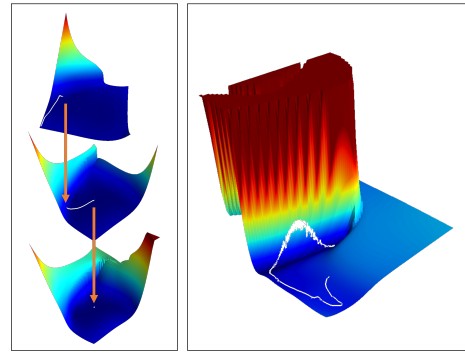

Figure 1: Optimization trajectories for our homotopy method **(left)** and vanilla gradient descent **(right)**. While convention training meanders on the pathological loss landscape, our method provides a series of relaxed landscapes that effectively guide the optimizer to the loss minimum.

In this work, we focus on Neural Ordinary Differential Equations (NeuralODEs) [11]. These models are powerful tools in modeling natural phenomena, bridging the expressibility and flexibility of neural networks with the de facto mathematical language of the physical sciences. This has led to researchers amalgamating NeuralODEs with partial information about the governing equation to produce "grey-box" dynamics models [55, 71], and endowing NeuralODEs with mathematical structures that the system must satisfy [22, 18]. Despite their conceptual elegance, training NeuralODEs tend to result in long training times and sub-optimal results, a problem that is further exacerbated as the length of the training data grows [20, 17]. Similar to the case of RNNs, methods have been proposed to tackle the problem involve placing either strong [12, 25], or semi-strong constraints [17, 30] to the functional form the NeuralODE can take - something the underlying governing equation does not guarantee satisfying.

**Contributions**   We introduce a method to accurately train NeuralODEs on long time series data by leveraging tools from the chaos and mathematical optimization literature: synchronization and homotopy optimization [1, 15]. Through loss landscape analyses, we show that longer training data lead to more irregular loss landscapes, which in turn lead to deteriorating performances of the trained models. We show that synchronizing NeuralODE dynamics with the data can smooth the loss landscape, on which homotopy optimization can be applied to enhance training. Through performance benchmarks, we show that not only does our model far outperform conventional gradient descent training, it also surpasses multiple shooting in terms of convergence speed and extrapolation accuracy.

## 2   Preliminaries

### 2.1   Neural ordinary differential equations

A NeuralODE [11] is a model of the form,

$$\frac{d\mathbf{u}}{dt} = \mathbf{U}(t, \mathbf{u}; \boldsymbol{\theta}), \quad \mathbf{u}(t_0) = \mathbf{u}_0. \tag{1}$$

where $\mathbf{u}_0 \in \mathbb{R}^n$ is the initial condition or input given to the model, and $\mathbf{U}(...; \boldsymbol{\theta}) : \mathbb{R} \times \mathbb{R}^n \to \mathbb{R}^n$ is a neural network with parameters $\boldsymbol{\theta} \in \mathbb{R}^m$ that governs the dynamics of the model state $\mathbf{u}(t) \in \mathbb{R}^n$. While this model can be used as a input-output mapping[1], we are interested in the problem of training

---

[1]In this case, NeuralODEs become the continuous analog of residual networks [11].

NeuralODEs on time series data to learn the underlying governing equation and forecast future dynamics.

Given an monotonically increasing sequence of time points and the corresponding measurements $\{t^{(i)}, \hat{\mathbf{u}}^{(i)}\}_{i=0}^N$, NeuralODE training starts with using an ordinary differential equation (ODE) solver to numerically integrate Equation (1) to obtain the model state $\mathbf{u}$ at given time points:

$$\mathbf{u}^{(i)}(\boldsymbol{\theta}) = \mathbf{u}^{(0)}(\boldsymbol{\theta}) + \int_{t^{(0)}}^{t^{(i)}} \mathbf{U}(t, \mathbf{u}; \boldsymbol{\theta}) dt = \text{ODESolve}\left(\mathbf{U}(t, \mathbf{u}; \boldsymbol{\theta}), t^{(i)}, \mathbf{u}_0\right). \tag{2}$$

Afterwards, gradient descent is performed on the loss function $\mathcal{L}(\boldsymbol{\theta}) = \frac{1}{N+1} \sum_i \left|\mathbf{u}^{(i)}(\boldsymbol{\theta}) - \hat{\mathbf{u}}^{(i)}\right|_2^2$ to arrive at the optimal parameters $\boldsymbol{\theta}^*$.

## 2.2 Homotopy optimization method

In topology, *homotopy* refers to the continuous deformation of one function into another. Motivated by this concept, to find the minimizer $\boldsymbol{\theta}^* \in \mathbb{R}^m$ of a complicated function $\mathcal{F}(\boldsymbol{\theta})$, the homotopy optimization method[3] [67, 68, 3, 15] introduces an alternative objective function

$$\mathcal{L}(\boldsymbol{\theta}, \lambda) = \begin{cases} \mathcal{G}(\boldsymbol{\theta}), & if \quad \lambda = 1 \\ \mathcal{F}(\boldsymbol{\theta}), & if \quad \lambda = 0 \end{cases} \tag{3}$$

where $\mathcal{G}(\boldsymbol{\theta})$ is an auxillary function whose minimum is easily found, and $\mathcal{L}(\boldsymbol{\theta}, \lambda) : \mathbb{R}^m \times \mathbb{R} \to \mathbb{R}$ smoothly interpolates between $\mathcal{G}$ and $\mathcal{F}$ as the homotopy parameter $\lambda$ varies from 1 to 0. While a commonly used construction is the convex homotopy $\mathcal{L}(\boldsymbol{\theta}, \lambda) = (1 - \lambda)\mathcal{F}(\boldsymbol{\theta}) + \lambda\mathcal{G}(\boldsymbol{\theta})$, more problem-specific forms can be employed.

Homotopy optimization shines in cases where the target function $\mathcal{F}$ is non-convex and has a complicated landscape riddled with local minima. Similarly to simulated annealing [33], one starts out with a relaxed version of the more complicated problem at hand and finds a series of approximate solutions while slowly morphing the relaxed problem back into its original non-trivial form. This allows the optimization process to not get stuck in spurious sharp valleys and accurately converge to the minimum of interest.

## 2.3 Synchronization of dynamical systems

Here, we briefly summarize the mathematical phenomenon of synchronization, as described in the literature [51, 28, 1, 49]. In later sections, we will show that synchronization can be used mitigate the difficulties arising in NeuralODE training. Consider two states $\hat{\mathbf{u}}, \mathbf{u} \in \mathbb{R}^n$, each evolving via

$$\frac{d\hat{\mathbf{u}}}{dt} = \hat{\mathbf{U}}(t, \hat{\mathbf{u}}; \hat{\boldsymbol{\theta}}), \quad \frac{d\mathbf{u}}{dt} = \mathbf{U}(t, \mathbf{u}; \boldsymbol{\theta}) - \mathbf{K}(\mathbf{u} - \hat{\mathbf{u}}). \tag{4}$$

Here, $\hat{\mathbf{U}}, \mathbf{U} : \mathbb{R} \times \mathbb{R}^n \to \mathbb{R}^n$ are the dynamics of the two systems parameterized by $\hat{\boldsymbol{\theta}}, \boldsymbol{\theta}$, respectively, and $\mathbf{K} = diag(k_1, ..., k_n)$, is the diagonal coupling matrix between the two systems. The two systems are said to synchronize if the error between $\hat{\mathbf{u}}(t)$ and $\mathbf{u}(t)$ vanishes with increasing time.

**Theorem 1** (Synchronization). *Assuming $\mathbf{U} = \hat{\mathbf{U}}$, $\boldsymbol{\theta} \approx \hat{\boldsymbol{\theta}}$, and $\mathbf{u}(t)$ in the vicinity of $\hat{\mathbf{u}}(t)$, the elements of $\mathbf{K}$ can be chosen so that the two states synchronize: that is, for error dynamics $\boldsymbol{\xi}(t) = \mathbf{u}(t) - \hat{\mathbf{u}}(t)$, $\|\boldsymbol{\xi}(t)\|_2 = \|\mathbf{u}(t) - \hat{\mathbf{u}}(t)\|_2 \to 0$ as $t \to \infty$.*

*Proof.* By subtracting the two state equations, the error, up to first order terms, can be shown to follow:

$$\frac{d\boldsymbol{\xi}(t)}{dt} = \frac{d\mathbf{u}(t)}{dt} - \frac{d\hat{\mathbf{u}}(t)}{dt} = \left[\left.\frac{\partial \mathbf{U}(\mathbf{t}, \mathbf{u}; \boldsymbol{\theta})}{\partial \mathbf{u}}\right|_{\mathbf{u}=\hat{\mathbf{u}}} - \mathbf{K}\right] \boldsymbol{\xi}(t) = \mathbf{A}\boldsymbol{\xi}(t). \tag{5}$$

Increasing the elements of $\mathbf{K}$ causes the matrix $\mathbf{A}$ to become negative definite —which in turn, causes the largest Lyapunov exponent of the system,

$$\Lambda := \lim_{t \to \infty} \frac{1}{t} \log \frac{\|\boldsymbol{\xi}(t)\|}{\|\boldsymbol{\xi}(0)\|}. \tag{6}$$

---

[2]It is also possible to use other types of losses, such as the L1 loss used in [18, 31].

[3]In the literature, these methods are also referred to as "continuation methods"

to also become negative as well. Then $||\boldsymbol{\xi}(t)||_2 = ||\mathbf{u}(t) - \hat{\mathbf{u}}(t)||_2 \to 0$ as $t \to \infty$, which means the dynamics of $\mathbf{u}$ and $\hat{\mathbf{u}}$ synchronize with increasing time regardless of the parameter mismatch. $\square$

**Remark.** *Our assumption of* $\mathbf{U} = \hat{\mathbf{U}}$, *that the functional form of the data dynamics is the same as the equation to be fitted can also be thought to hold for NeuralODE training. This is because if the model is sufficiently expressive, universal approximation theorem will allow the model to match the data equation for some choice of model parameter.*

### 2.3.1 Constructing homotopy with synchronization

To combine homotopy optimization with synchronization, we slightly modify the coupling term of Equation (4) by multiplying it with the homotopy parameter $\lambda$:

$$\frac{d\mathbf{u}}{dt} = \mathbf{U}(t, \mathbf{u}; \boldsymbol{\theta}) - \lambda\mathbf{K}(\mathbf{u} - \hat{\mathbf{u}}). \tag{7}$$

With this modification, homotopy optimization is introduced to the problem as follows. When $\lambda = 1$ and the coupling matrix $\mathbf{K}$ has sufficiently large elements, synchronization occurs and the NeualODE prediction $\mathbf{u}(t)$ starts to converge to the data trajectory $\hat{\mathbf{u}}(t)$ for large $t$. As we will confirm in Section 4, this causes the resulting loss function $\mathcal{L}(\boldsymbol{\theta}, 1)$ to become well-behaved, serving the role of the auxillary function $\mathcal{G}$ in Equation (3). When $\lambda = 0$, the coupled Equation (4) reduces to the original model form of Equation (1). Accordingly, the corresponding loss function $\mathcal{L}(\boldsymbol{\theta}) = \mathcal{L}(\boldsymbol{\theta}, 0)$ is the complicated loss function we need to ultimately minimize. Therefore, starting with $\lambda = 1$ and successively decreasing its value to 0, all the while optimizing for the coupled loss function $\mathcal{L}(\boldsymbol{\theta}, \lambda)$ allows one to leverage the well-behaved loss function landscape from synchronization while being able to properly uncover the system parameters [66, 58].

## 3 Related works

**Synchronization in ODE parameter estimation**   After the discovery of synchronization in chaotic systems by Pecora & Carroll [47, 48], methods to leverage this phenomenon to identify unknown ODE parameters were first proposed by Parlitz [44], and later refined by Abarbanel et al. [2, 1]. Parameter identification in these earlier works would proceed by either designing an auxillary dynamics for the ODE parameters [44, 40], or including a regularization term on the coupling strength in the loss function [53, 1]. The idea of applying homotopy optimization to this problem was pioneered by Vyasarayani et al. [65, 66] and further expanded upon by later works [58, 63, 52].

**Homotopy methods in machine learning**   As homotopy methods can outperform gradient descent for complex optimization problems, it has been used in diverse fields, including Gaussian homotopy for non-convex optimization [27] and training feedforward networks [13, 24, 10]. Curriculum learning [6] is another example of homotopy optimization, with the each task in the curriculum corresponding to a particular value of the homotopy parameter as it is decreased to 0. Not to be confused with the method discussed in this paper is the homotopy perturbation technique [26] used in the dynamical systems literature, which is a method to obtain analytical solutions to nonlinear differential equations.

**Improving the training of RNNs and NeuralODEs**   As previously mentioned, many recent approaches to improve RNN and NeuralODE training involve constraining the model expressivity. The limitations of this strategy is investigated in [57], where it is demonstrated that such "stabilized" models cannot learn oscillatory or highly complex data that does not match the built-in restrictions. A recent development with a similar spirit as ours is Mikhaeil et al. [41], where the authors developed an architecture-agnostic training method for RNNs by revisiting the classical RNN training strategy of teacher forcing [60] and refining it via the chaos theory concept of Lyapunov exponents. Other aspects of NeuralODE training, such as better gradient calculation, have been studied to improve NeuralODE performance [30, 72, 39, 31]. Such methods are fully compatible with our proposed work, and we showcase an example in Appendix G, by changing the gradient calculation scheme to the "symplectic-adjoint-method" from [39].

# 4 Loss landscapes in learning ODEs

## 4.1 Complicated loss landscapes in ODE parameter estimation

NeuralODEs suffer from the same training difficulties of ODE fitting and RNNs. Here we illustrate that the underlying cause for this difficulty is also identical by analyzing the loss landscape of NeuralODEs for increasing lengths of training data. The data were generated from the periodic Lotka-Volterra system[4] and the landscapes were visualized by calculating the two dominant eigenvectors of the loss Hessian, then perturbing the model parameter along those directions [69]. We also provide analogous analyses for ODE parameter estimation on the same system in Appendix B to further emphasize the similarities between the problem domains.

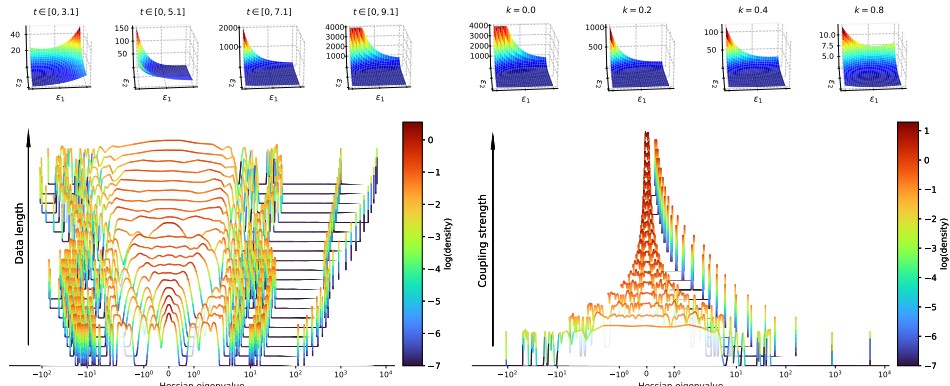

Figure 2: Irregular loss landscape in NeuralODE training. **(Upper left)** Loss landscape and **(Lower left)** eigenvalue spectrum of the loss Hessian for increasing lengths of train data. **(Upper right)** Loss landscape and **(Lower right)** eigenvalue spectrum of the loss Hessian with increasing coupling strength. For clarity, loss values were clipped above at 4000. All Hessian related information was calculated using the `PyHessian` package [69].

From Figure 2 (upper left), we find that the loss landscape is well-behaved for short data, but becomes pathological as the data lengthens. In fact, the landscape for $t \in [0, 9.1]$ features a stiff cliff, which is known to cause exploding gradients in RNN training [14, 45]. Indeed, this picture is also confirmed by an alternative principal component analysis-based visualization of the loss landscape and the optimization path [35][5] (Figure 1, right). We also observe that the overall magnitude of the loss increases with data length. This is a direct consequence of model dynamics $\mathbf{u}(t)$ being independent of the data dynamics $\hat{\mathbf{u}}(t)$ and therefore tending to diverge away with time.

To further confirm our findings and compensate for the fact that our visualizations are low dimensional projections of the actual loss landscape, we computed the eigenvalue spectrum of the loss function hessian for increasing lengths of training data (Figure 2, lower left). The computed spectrums show that hessian eigenvalues spread away from zero with increasing length of training data, signalling that the loss morphs from a relatively flat to a more corrugated landscape. We also observe large positive eigenvalue outlier peaks, whose values also rapidly increase with the length of training data. As such large outliers have been report to slow down training [19], this also provides additional insights as to why NeuralODE training stagnates on long time series data.

## 4.2 Loss landscape of synchronized dynamics

With coupling, the loss function $\mathcal{L}$ between the model predictions and the data now becomes a function of both the model parameters $\boldsymbol{\theta}$ and the coupling strength $k$. Increasing the coupling strength $k$ effectively acts as shortening the data because increased coupling causes the two trajectories to start synchronizing earlier in time, after which the error between the two will diminish and contribute negligibly to the total loss. Therefore, synchronizing the model and the data generating equations leads to a more favorable loss landscape that gradient descent based optimizers will not struggle on.

---

[4]For more details, see Section 6

[5]For details on the visualization method, see Section B of the supplementary of the cited reference.

Once again, we confirm this point by visualizing the loss landscape and the Hessian eigenvalue spectrum for a long time series dataset while increasing the strength of the coupling. From Figure 2 (upper right) we find that with increasing coupling strength, the steep cliff and the flat basin of the original loss surface are tamed into a well-behaved slope. This is reflected in the eigenvalue spectrum as well (Figure 2, lower right), with the all of the spectrum gathering near zero as the coupling is turned up. In fact, when the coupling strength is increased too much, we find that all the eigenvalues bunch together at zero, meaning that the loss function becomes completely flat. Qualitatively, this corresponds to the case where the coupling completely dominates the synchronized dynamics and forces the model predictions onto the data regardless of the model parameters.

## 5 Homotopy optimization for NeuralODE training

Having established that applying synchronization between the model and data can tame the irregular loss landscape during training, we now turn towards adapting the synchronization-homotopy approach of Section 2.3.1 to a NeuralODE setting. In doing so, there are couple details that needs addressing to arrive at a specific implementations. We discuss such points below.

### 5.1 Constructing the coupling term

The coupling term for synchronization $-\lambda\mathbf{K}(\mathbf{u} - \hat{\mathbf{u}})$ in Equation (7) depends on the data dynamics $\hat{\mathbf{u}}$, whose values are only available at discrete timestamps $\{t^{(i)}\}_{i=0}^N$. This is at a disaccord with the ODE solvers used during training, which require evaluating the model at intermediate time points as well. Therefore, to construct a coupling term defined for arbitrary time, we used a cubic smoothing spline [21] of the data points to supply the $\hat{\mathbf{u}}$ values at unobserved times. Smoothing was chosen over interpolation to increase the robustness of our approach to measurement noise.

Note that cubic spline is not the only solution - even defining the coupling term as a sequence of impulses is possible, only applying it where it is defined, and keeping it zero at other times. This is an interesting avenue of future research, as this scheme can be shown [53] to be mathematically equivalent to teacher forcing [60].

### 5.2 A little more on the coupling matrix

In general, the coupling matrix $\mathbf{K}$ can be any $n \times n$ matrix, provided it renders the Jacobian of the error dynamics (matrix $\mathbf{A}$ in Equation (6)) to be negative definite along the data trajectory $\hat{\mathbf{u}}(t)$. However, naively following this approach leads to $n^2$ hyperparameters just for the coupling matrix, which can quickly get out of hand for high dimensional systems. To circumvent this issue, in this work we have used a coupling matrix of the form $\mathbf{K} = k\mathbf{I}_n$, where $\mathbf{I}_n$ is the $n \times n$ identity matrix, to arrive at single scalar hyperparameter characterizing coupling strength.

### 5.3 Scheduling the homotopy parameter

The homotopy parameter $\lambda$ can be decreased in various different ways, the simplest approach being constant steps. In our study, we instead used power law decrements of the form $\{\Delta\lambda^{(j)}\}_{j=0}^{n_{step}-1} = \kappa^j / \sum_{j=0}^{n_{step}-1} \kappa^j$ where $\Delta\lambda^{(j)}$ denotes the $j$-th decrement and $n_{step}$ is number of homotopy steps. This was inspired from our observations of the synchronized loss landscape (Figure 2, right), where the overall shape change as well as the width of the eigenvalue spectrum seemed be a superlinear function of the coupling strength.

### 5.4 Hyperparameter overview

Our final implementation of the homotopy training algorithm has five hyperparameters, as described below.

**Coupling strength ($k$)**  This determines how trivial the auxillary function for the homotopy optimization will be. Too small, and even the auxillary function will have a jagged landscape; too large, and the initial auxillary function will become flat (Figures 2 and 11, left panel, $k = 1.0, 1.5$) resulting in very slow parameter updates. We find good choices of $k$ tend to be comparable to the scale of the measurement values.

---

**Algorithm 1** Homotopy-based NeuralODE training

---

**Input:** Data $\{t^{(i)}, \hat{\mathbf{u}}^{(i)}\}_{i=0}^{N}$, untrained model $\mathbf{U}(...; \boldsymbol{\theta})$

  **Hyperparameters:** Coupling strength($k$), homotopy parameter decrement ratio($\kappa$), number of homotopy steps($s$), epochs per homotopy step($n_{epoch}$), learning rate($\eta$)

  $\hat{\mathbf{u}}(t) = CubicSmoothingSpline(\{t^{(i)}, \hat{\mathbf{u}}^{(i)}\}_{i=0}^{N})$            ▷ Construct data interpolant

  $\lambda \leftarrow 1$                                                   ▷ Initialize homotopy parameter

  **for** $s$ in $0 \ldots n_{step} - 1$ **do**

      **for** $epoch$ in $0 \ldots n_{epoch} - 1$ **do**

          $\{\mathbf{u}^{(i)}\}_{i=0}^{N} = ODESolve(\dot{\mathbf{u}} = \mathbf{U}(t, \mathbf{u}; \boldsymbol{\theta}) - \lambda k \mathbf{I}_n(\mathbf{u} - \hat{\mathbf{u}}(t)))$ ▷ Solve coupled dynamics

          $\{\tilde{\mathbf{u}}^{(i)}\}_{i=0}^{N} = ODESolve(\dot{\tilde{\mathbf{u}}} = \mathbf{U}(t, \tilde{\mathbf{u}}; \boldsymbol{\theta}))$           ▷ Also solve uncoupled dynamics

          $MSE(\boldsymbol{\theta}) = \frac{1}{N+1}\sum_i \left|\tilde{\mathbf{u}}^{(i)}(\boldsymbol{\theta}) - \hat{\mathbf{u}}^{(i)}\right|^2$

          **if** $MSE \leq MSE_{best}$ **then**

              $MSE_{best} \leftarrow MSE$       ▷ Model performance is gauged by uncoupled dynamics error

              $\boldsymbol{\theta}^* \leftarrow \boldsymbol{\theta}$                          ▷ Checkpoint best model parameters

          **end if**

          $\mathcal{L}(\boldsymbol{\theta}) = \frac{1}{N+1}\sum_i \left|\mathbf{u}^{(i)}(\boldsymbol{\theta}) - \hat{\mathbf{u}}^{(i)}\right|^2$ ▷ Loss function is calculated using coupled dynamics

          $\boldsymbol{\theta} \leftarrow OptimizerStep(\eta, \boldsymbol{\theta}, \nabla_{\boldsymbol{\theta}}\mathcal{L})$

      **end for**

      $\lambda \leftarrow \lambda - \kappa^s / \sum_{s=0}^{n_{steps}-1} \kappa^s$      ▷ Decrease homotopy parameter with power law decrements

  **end for**

**Output:** Trained model parameters $\boldsymbol{\theta}^*$, $\lambda = 0$

---

**Homotopy parameter decrement ratio ($\kappa$)** This determines how the homotopy parameter $\lambda$ is decremented for each step. Values close to 1 cause $\lambda$ to decrease in nearly equal decrements, whereas smaller values cause a large decrease of $\lambda$ in the earlier parts of the training, followed by subtler decrements later on. We empirically find that $\kappa$ values near 0.6 tend to work well.

**Number of homotopy steps ($s$)** This determines how many relaxed problem the optimization process will pass through to get to the final solution. Similar to scheduling the temperature in simulated annealing, fewer steps results in the model becoming stuck in a sharp local minima, and too many steps makes the optimization process unnecessarily long. We find using values in the range of 6-8 or slightly larger values for more complex systems yields satisfactory results.

**Epochs per homotopy step ($n_{epoch}$)** This determines how long the model will train on a given homotopy parameter value $\lambda$. Too small, and the model lacks the time to properly converge on the loss function; too large, and the model overfits on the simpler loss landscape of $\lambda \neq 0$, resulting in a reduced final performance when $\lambda = 0$. We find for simpler monotonic or periodic systems, values of 100-150 work well; for more irregular systems, 200-300 are suitable.

**Learning rate ($\eta$)** This is as same as in conventional NeuralODE training. We found values in the range of 0.002-0.1 to be adequate for our experiments.

## 6 Experiments

To benchmark our homotopy training algorithm against existing methods, we trained multiple NeuralODE models across time-series datasets of differing complexities. The performance metrics we used in our experiments were mean squared error on the training dataset and on model rollouts of 100% the training data length, which from now on we refer to as interpolation and extrapolation MSE, respectively. For each training method, the hyperparameters were determined through a grid search performed prior to the main experiments. All experiments were repeated three times with different random seeds, and we report the corresponding means and standard errors whenever applicable. For conciseness, we outline the datasets, models and the baseline methods used in the subsequent sections, and refer interested readers to Appendices E and F for extensive details on data generation, model configuration and hyperparameter selection. We provide all of the code for this paper in `https://github.com/Jhko725/NeuralODEHomotopy`.

## 6.1 Datasets and model architectures

**Lotka-Volterra system**   The Lotka-Volterra system is a simplified model of prey and predator populations [43]. The system dynamics are periodic, meaning even models trained on short trajectories should have strong extrapolation properties, provided the training data contains a full period of oscillation. We considered two types of models for this system: a "black-box" model of Equation (1), and a "gray-box" model used in [54] given by $\frac{dx}{dt} = \alpha x + U_1(x, y; \boldsymbol{\theta}_1)$, $\frac{dy}{dt} = -\gamma y + U_2(x, y; \boldsymbol{\theta}_2)$.

**Double pendulum**   For this system, we used the real-world measurement data from Schmidt & Lipson [59]. While the double pendulum system can exhibit chaotic behavior, we used the portion of the data corresponding to the aperiodic regime. We trained two types of models on this data: a black-box model and a model with second-order structure. Our motivation for including the latter model stems from the ubiquity of second-order systems in physics and from [23], which reports these models can match the performance of Hamiltonian neural networks.

**Lorenz system**   The Lorenz system displays highly chaotic behavior, and serves as a stress test for time series prediction tasks. For this system, we only employed a black-box model.

## 6.2 Baselines

As the focus of this paper is to effectively train NeuralODEs without employing any specialized model design or constraining model expressivity, this was also the criteria by which we chose the baseline training method for our experiments. Two training methods were chosen as our baseline: vanilla gradient descent, which refers to the standard method of training NeuralODEs, and multiple-shooting method. As the latter also stems from the ODE parameter estimation literature, and its inner working relevant to the discussion of our results, we briefly describe the multiple-shooting algorithm before discussing our results.

**Multiple shooting**   The muliple-shooting algorithm [62, 8, 5, 64] circumvents the difficulty of fitting models on long sequences by dividing the training data into shorter segments and solving multiple initial value problems. To keep the predictions for different segments in sync, the loss function is augmented with a continuity enforcing term. Unlike synchronization and homotopy, which we employ in NeuralODE training for the first time, works applying multiple shooting to NeuralODEs are scant but do exist [54, 38, 61][6]. As the three previous references differ in their implementations, we chose the scheme of [54] for its simplicity and implemented a `PyTorch` adaptation to use for our experiments. Further details about our multiple-shooting implementation can be found in Appendix D.

# 7   Results

## 7.1   Training NeuralODEs on time series data of increasing lengths

In the previous sections, we found that longer training data leads to a more irregular loss landscape. To confirm that this does translate to deteriorated optimization, we trained NeuralODEs on Lotka-Volterra system trajectories of different lengths. Inspecting the results (Figure 3, first panel) we find while all training methods show low interpolation MSE for shortest data, this is due to overfitting, as evidenced by the diverging extrapolation MSE[7]. As the length of the data grows and encompasses a full oscillation period, we find that the model performances from homotopy and multiple-shooting stabilize for both interpolation and extrapolation. This is clearly not the case for the vanilla gradient descent, whose model performances degrade exponentially with as data length increases.

**Model capacity is not the problem**   The reason NeuralODEs struggle on long sequences can easily be attributed to insufficient model capacity. We demonstrate this is not the case by repeating the previous experiment for the mid-length data while decreasing the model size used (Figure 3, second panel). In the case of vanilla gradient descent, the model size is directly correlated to its performance,

---

[6]While not explicitly mentioned in the paper, `DiffEqFlux.jl` library has an example tutorial using this technique to train NeuralODEs.

[7]Indeed, plots of the model prediction trajectory in Appendix G.3 also confirm this fact.

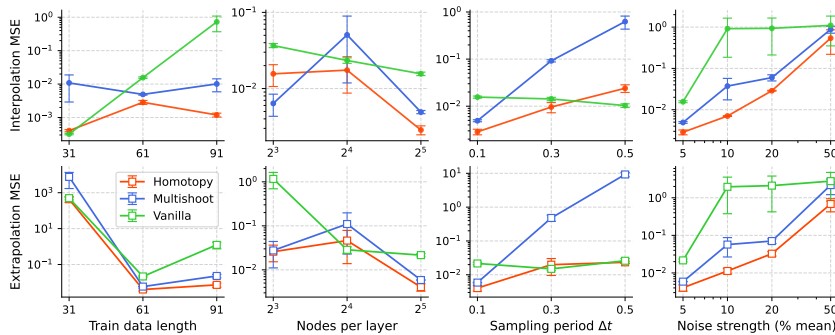

Figure 3: Results for the Lotka-Volterra system. **(First)** Interpolation and extrapolation errors for increasing data length. **(Second)** Errors for the mid-length data with decreasing model capacity. **(Third)** Errors for fixed time interval but with increasing data sparsity. **(Fourth)** Errors for mid-length data with increasing noise.

lending the claim some support. However, we find that both homotopy and multiple shooting methods can effectively train much smaller models to match the accuracy of a larger model trained with vanilla gradient descent. This signifies that the cause of the problem is not the capacity itself, but rather the inability of training algorithms to fully capitalize on the model capacity.

**Robustness against data degradation** We performed an additional set of experiments to evaluate the performances of different training methods against increasing data sparsity and noise. This is particularly of interest for our homotopy training as we employ smoothing cubic splines (Section 5.1) whose reliability can be adversely affected by the data quality. From the results (Figure 3, third, fourth panels) we find this is not the case and that that our algorithm performs well even for degraded data. One unexpected result is the drastic underperformance of the multiple shooting method with increased sampling period, which is a side effect from the fact we held the time interval fixed for this experiment. Due to the space constraint, we continue this discussion in Appendix G.3.

## 7.2 Performance benchmarks of the training algorithms

Having confirmed that both the homotopy and multiple-shooting methods successfully mitigate the NeuralODE training problem, we now present a comprehensive benchmark in Figure 4.

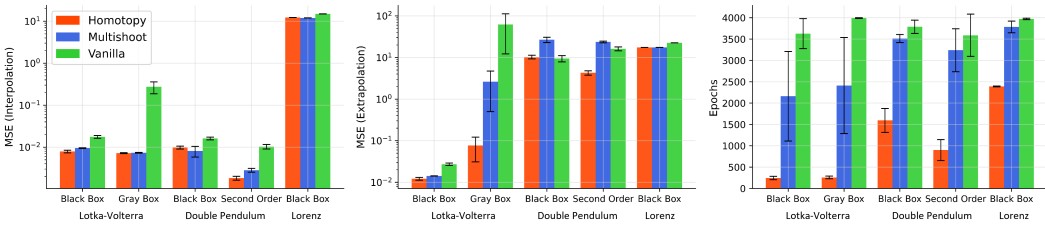

Figure 4: Performance benchmarks. **(Left)** Interpolation and **(Center)** extrapolation MSEs for each training method. **(Right)** Epochs where the minimum interpolation MSE was achieved. For all plots, the bar and the error bars denote the mean and standard errors across three runs.

From the results, it is clear both our homotopy method and multiple shooting can both boost the interpolation performance of NeuralODEs, with our method being slightly more performant. Inspecting the extrapolation performance, however, we do find a clear performance gap between our method and multiple shooting, with the latter even performing worse than vanilla gradient descent for the double pendulum dataset (Figure 4, center, Figure 5, left). This underperformance of multiple shooting for extrapolation tasks can be attributed to the nature of the algorithm: during training time, multiple shooting integrates NeuralODEs on shorter, subdivided time intervals, whereas during inference time the NeuralODE needs to be integrated for the entire data time points and then some more for extrapolation. This disparity between training and inference render models trained by

multiple shooting to be less likely to compensate for error accumulation in long term predictions, resulting in poor extrapolation performance.

Another stark difference between our homotopy training and the other methods is in the number of epochs required to arrive at minimum interpolation MSE (Figure 4, right panel). While the exact amount of speedup is variable and seems to depend on the dataset difficulty, we find a 90%-40% reduction compared to vanilla training, highlighting the effectiveness of our approach. One unexpected observation from our experiments was that the epoch at which minimum interpolation MSE epoch was achieved did not always correspond to the homotopy parameter $\lambda$ being zero (Appendix G.2). While seemingly counterintuitive, this is in line with the literature, where nonzero homotopy parameter has resulted in better generalization performances [36, 24].

**What makes homotopy training effective: inspecting training dynamics**  To obtain a complementary insight to the NeuralODE training dynamics and relate the effectiveness of our method with the properties of the loss landscape, we calculated the trace of the loss hessian during training (Figure 5, right panel). Small, positive trace values indicate more favorable training dynamics [69], and we find that our method results in the smallest trace, followed by multiple shooting then vanilla gradient descent. The fact that the hessian trace maintains a low value throughout the entire homotopy training, even as the loss surface is morphed back to its unsynchronized, irregular state, also confirms that our homotopy schedule works well and managed to gradually guide the optimizer into a well performing loss minimum.

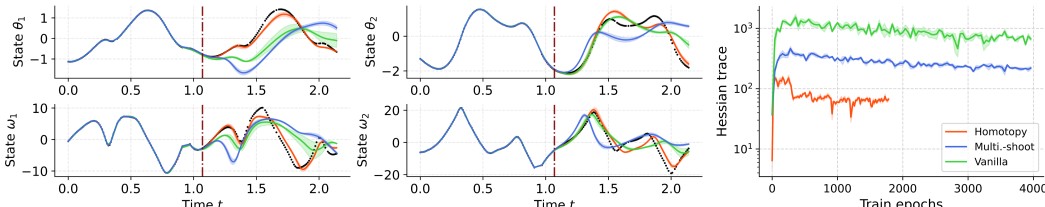

Figure 5: Second-order NeuralODE training results for the double pendulum dataset. **(Left)** Predicted trajectories. Lines and bands correspond to mean and standard errors for three training runs and the dashed line indicate the extrapolation start time. **(Right)** Estimation of the hessian trace during training. The homotopy curve stops early due to the difference in the number of maximum training curves used.

# 8    Limitations and discussion

In this paper, we adapted the concepts of synchronization and homotopy optimization for the first time in the NeuralODE literature to stabilize NeuralODE training on long time-series data. Through loss landscape analysis, we established that the cause of the training problem is identical between differential equations, RNNs and NeuralODEs and that synchronization can effectively tame the irregular landscape. We demonstrated that our homotopy method can successfully train NeuralODE models with strong interpolation and extrapolation accuracies.

Note that our work serves as an initial confirmation that the methods developed in the ODE parameter estimation literature can be adapted to be competitive in the NeuralODE setting. As such, there are multiple opportunities for future research, including theoretical studies on the convergence properties of our algorithm and further testing out ideas in the synchronization literature, such as alternative parametrization for the coupling matrix [28], or incorporating the homotopy parameter into the loss function as a regularization term [1, 58]. Extending our algorithm to high-dimensional systems is another important direction, which we believe will be possible through the use of latent ODE models [70]. Nevertheless, our work hold merit in that it lays the groundwork upon which ideas from ODE parameter estimation, control theory, and RNN/NeuralODE training can be assimilated, which we believe will facilliate further research connecting these domains.

**Author Contributions**

J. Ko and H. Koh contributed equally to the paper.

**Acknowledgments**

This work was supported by grants from the National Research Foundation of Korea (No. 2016R1A3B1908660) to W. Jhe.

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

# A    Additional figures

In this section, we present some of the figures that were excluded from the main text for brevity.

Figure 6 below highlights the efficiency of our training approach with respect to the number of training epochs by plotting the trajectory predictions of the second-order models during training.

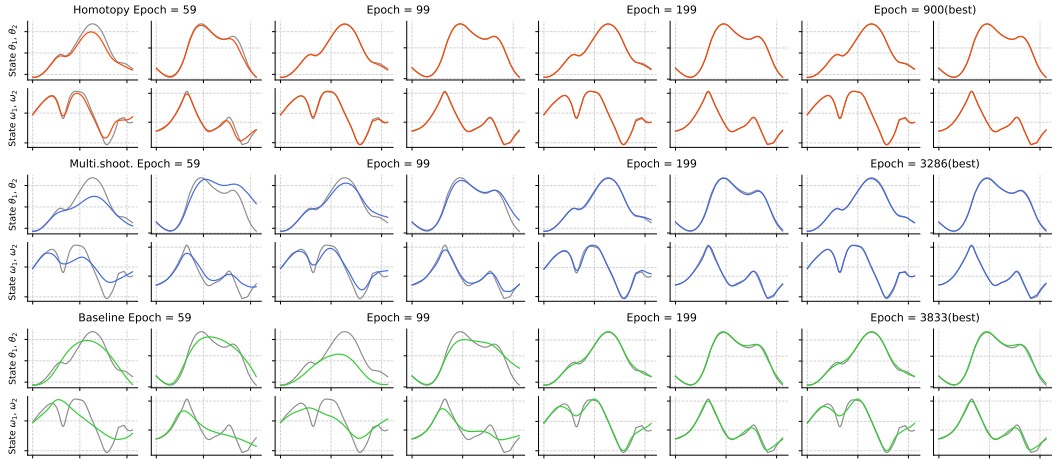

Figure 6: Second-order model predictions for the double pendulum data during training. Blue and orange lines indicate data and model predictions respectively. **(Top)** Results from our homotopy approach. **(Middle)** Results from multiple shooting. **(Bottom)** Results from vanilla gradient descent.

Figures 7 to 9 show model prediction trajectories corresponding to our benchmark results of Figure 4. One intriguing point is that even though both the gray-box model (Lotka-Volterra system, Figure 7) and the second-order model (double pendulum, Figure 8) contain more prior information about their respective systems, this does not necessarily translate to better predictive capabilities if an effective training method is not used. This suggests that introducing physics into the learning problem can obfuscate optimization, something that has been reported for physics-informed neural networks [32]. It also highlights the effectiveness of our homotopy training algorithm, as our method can properly train such difficult-to-optimize models.

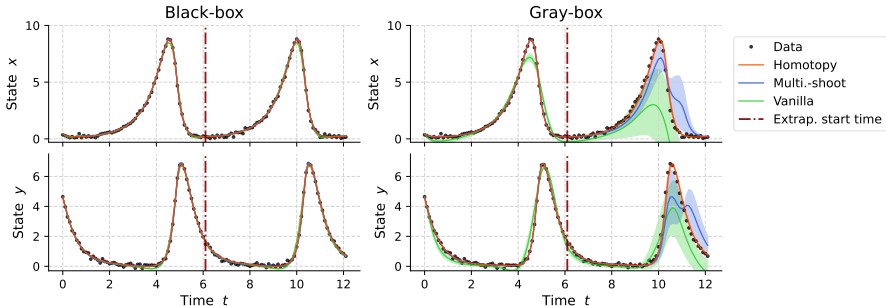

Figure 7: Trajectory predictions for the Lotka-Volterra system from models trained with different methods. Dashed vertical line indicates the start of extrapolation.

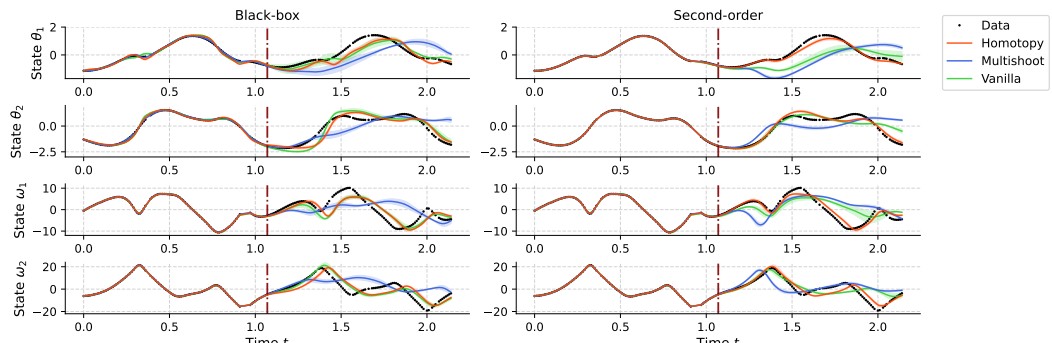

Figure 8: Trajectory predictions for the double pendulum from models trained with different methods. Dashed vertical line indicates the start of extrapolation.

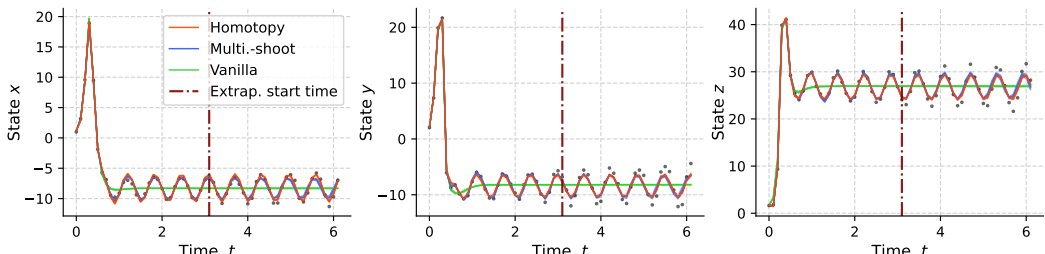

Figure 9: Trajectory predictions for the Lorenz system from models trained with different methods. Dashed vertical line indicates the start of extrapolation.

## B  Loss landscape and synchronization in ODE parameter estimation

Here, we illustrate the similarity between the ODE parameter estimation problem and the NeuralODE training problem by repeating analyses analogous to those of Section 4 in the setting of estimating the unknown coefficients of a given ordinary differential equation.

In the left and middle panels of Figure 10, we show the trajectories of the Lotka-Volterra and the Lorenz systems where a single parameter was perturbed. We can easily observe that the original trajectory and the perturbed trajectories evolve independently in time. Furthermore, while trajectories from the simpler Lotka-Volterra system retain the same periodic behavior and differ only in their amplitudes and phases, trajectories from the Lorenz system display much different characteristics with increasing time. This translates over to the shape of the loss landscape (Figure 10, right) with the loss for the Lotka-Volterra system only becoming steeper with increasing data length, whereas the loss for the Lorenz system displays more and more local minima.

Once synchronization is introduced to the perturbed trajectory, its dynamics tends to converge with the reference trajectory as time increases, with the rate of convergence increasing for larger values of the coupling strength $k$ (Figure 11, left and middle panels). In terms of the loss landscape (right panel), larger coupling does result in a smoother landscape, with excessive amounts of it resulting in a flat landscape that is also detrimental to effective learning. These results are a direct parallel to our observations in Figure 2 and provide additional justifications as to why techniques developed in the field of ODE parameter estimation also work so well on NeuralODE training.

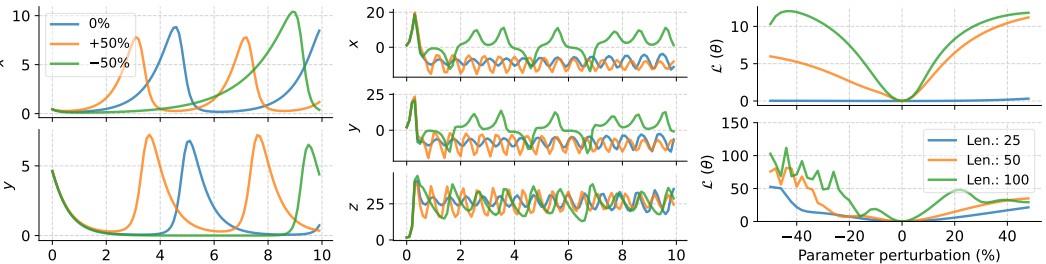

Figure 10: Dynamics of single parameter perturbed systems. The systems and the parameters used are described in Section 6. **(Left)** Solutions for the periodic Lotka-Volterra equations with perturbed $\alpha$ parameter. **(Middle)** Solutions for the chaotic Lorenz system with perturbed $\beta$ parameter. **(Right)** MSE loss function landscape for the Lotka-Volterra equations **(right, upper)** and Lorenz system **(right, lower)** for different lengths of time series data for the loss calculation.

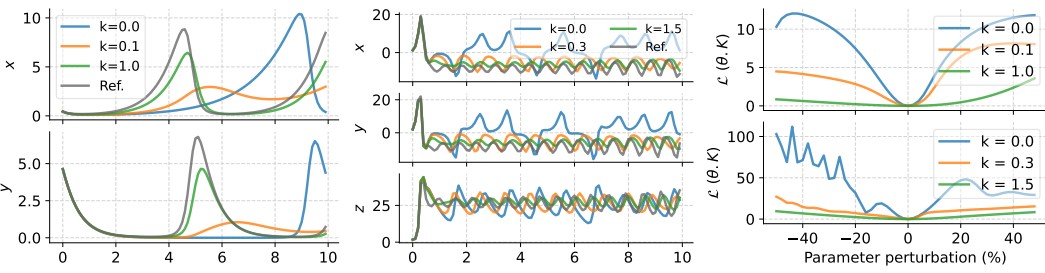

Figure 11: Dynamics of coupled systems with varying coupling strengths. The systems and the parameters used are identical to those of Figure 10. **(Left)** Results for the periodic Lotka-Volterra equations. **(Middle)** Results for the chaotic Lorenz system. **(Right)** Loss function landscape of the coupled systems with different coupling strengths.

## C    Further description of the homotopy optimization procedure

To implement homotopy optimization, one usually selects the number of discrete steps for optimization, $s$, as well as a series of positive decrement values for the homotopy parameter $\{\Delta\lambda^{(k)}\}_0^{s-1}$ that sum to 1. Afterwards, optimization starts with an initial $\lambda$ value of $\lambda^{(0)} = 1$, which gives $\mathcal{H}^{(0)}(\boldsymbol{\theta}) = \mathcal{G}(\boldsymbol{\theta})$. At each step, the objective function at the current iteration is minimized with respect to $\boldsymbol{\theta}$, using the output from the previous step $\boldsymbol{\theta}^{*(k-1)}$ is as the initial guess:

$$\mathcal{H}^{(k)}(\boldsymbol{\theta}) = \mathcal{H}(\boldsymbol{\theta}, \lambda = \lambda^{(k)}) \;\rightarrow\; \boldsymbol{\theta}^{*(k)} = \underset{\boldsymbol{\theta}}{\mathrm{argmin}}\, \mathcal{H}^{(k)}(\boldsymbol{\theta}) \tag{8}$$

Afterwards, $\lambda$ decremented to its next value, $\lambda^{(k+1)} = \lambda^{(k)} - \Delta\lambda^{(k)}$, and this iteration continues until the final step $s$ where $\lambda^{(s)} = 0$, $\mathcal{H}^{(s)}(\boldsymbol{\theta}) = \mathcal{F}(\boldsymbol{\theta})$, and the final minimizer $\boldsymbol{\theta}^{*(s)} = \boldsymbol{\theta}^*$ is the sought-after solution to the original problem $\mathcal{F}(\boldsymbol{\theta})$.

## D    Multiple shooting algorithm

Our implementation of the multiple shooting algorithm for training NeuralODEs closely mirrors the example code provided in the `DiffEqFlux.jl` package [54] of the Julia programming language ecosystem.

Given time series data $\{t_i, \hat{\mathbf{u}}_i\}_{i=0}^N$, multiple shooting method partitions the data time points into $m$ overlapping segments:

$$[t_0 = t_0^{(0)}, \dots, t_n^{(0)}], \dots, [t_0^{(m-1)}, \dots, t_n^{(m-1)} = t_N];\; t_n^{(i-1)} = t_0^{(i)},\, i \in 1 \dots m-1.$$

During training, the NeuralODE is solved for each time segment, resulting in $m$ segmented trajectories:

$$[\mathbf{u}_0^{(0)}, \ldots, \mathbf{u}_n^{(0)}], \ldots, [\mathbf{u}_0^{(m-1)}, \ldots, \mathbf{u}_n^{(m-1)}].$$

Taking into account the overlapping time point between segments, these trajectories can then all be concatenated to produce the full trajectory prediction:

$$[\mathbf{u}_0^{(0)}, \ldots, \mathbf{u}_{n-1}^{(0)}, \mathbf{u}_0^{(1)}, \ldots, \mathbf{u}_n^{(m-1)}] = [\mathbf{u}_0, \ldots \mathbf{u}_n].$$

From this, the data loss is defined identically to conventional NeuralODE training: $\mathcal{L}_{data}(\boldsymbol{\theta}) = \frac{1}{N+1} \sum_i |\mathbf{u}_i(\boldsymbol{\theta}) - \hat{\mathbf{u}}_i|^2$.

However, due to the segmented manner in which the above trajectory is generated, optimizing only over this quantity will result in a model that could generate good piecewise predictions, but is unable to generate a proper, continuous global trajectory. Therefore, to ensure the model can generate smooth trajectories, continuity constraints $\mathbf{u}_n^{(i-1)} = \mathbf{u}_0^{(i)}$, $(i \in 1 \ldots m-1)$ must be enforced. How this is achieved differs across the literature, and our implementation - following that of [54] - introduces a regularization term in the loss:

$$\mathcal{L}_{continuity}(\boldsymbol{\theta}) = \frac{1}{m} \sum_i \left| \mathbf{u}_n^{(i-1)} - \mathbf{u}_0^{(i)} \right|^2.$$

Finally, the train loss for the multiple shooting method is then defined as

$$\mathcal{L}(\boldsymbol{\theta}, \beta) = \mathcal{L}_{data}(\boldsymbol{\theta}) + \beta \cdot \mathcal{L}_{continuity}(\boldsymbol{\theta})$$

and is minimized using gradient descent, where $\beta$ is a hyperparameter that tunes the relative importance of the two terms during training.

# E  Experiment details

Our experiments were implemented in `PyTorch` [46], using the `pytorch-lightning` [16] framework. The `wandb` library [7] was used to keep track of all experiments as well as to perform hyperparameter sweeps. In all experiments, the AdamW optimizer [37] was used to minimize the respective loss functions for the vanilla and homotopy training. Each experiment was repeated using random seed values of 10, 20, and 30 to compute the prediction means and standard errors.

## E.1  Lotka-Volterra system

The Lotka-Volterra system is a simplified model of predator-prey dynamics given by,

$$\frac{dx}{dt} = \alpha x - \beta xy, \quad \frac{dy}{dt} = -\gamma y + \delta xy \tag{9}$$

where the parameters $\alpha, \beta, \gamma, \delta$ characterize interactions between the populations.

**Data preparation.**  Following the experimental design of Rackauckas et al. [55], we numerically integrate Equation (9) in the time interval $t \in [0, 6.1]$, $\Delta t = 0.1$ with the parameter values $\alpha, \beta, \gamma, \delta = 1.3, 0.9, 0.8, 1.8$ and initial conditions $x(0), y(0) = 0.44249296, 4.6280594$. Continuing with the recipe, Gaussian random noise with zero mean and standard deviations with magnitude of 5% of the mean of each trajectory was added to both states. For both data generation and NeuralODE prediction, integration was performed using the adaptive step size `dopri5` solver from the `torchdiffeq` package [11] with an absolute tolerance of 1e-9 and a relative tolerance of 1e-7.

For the experiments of Figure 3, training data were generated in a similar manner, but with specific parameters varied to match the corresponding experiments. Data varying train data length (Figure 3, left panel) were generated using time spans of $t \in [0, 3.1], [0, 6.1], [0.9.1]$ respectively with shared parameter values of $\Delta t = 0.1$ and noise amplitude of 5% of the mean. Data with differing sampling periods (Figure 3, third panel) were generated with a fixed time span of $t \in [0, 6.1]$, noise amplitude of 5% of the mean and sampling periods of $\Delta t = 0.1, 0.3, 0.5$. Data with different noise amplitudes (Figure 3, fourth panel) were generated using a time span of $t \in [0, 6.1]$, sampling period of $\Delta t = 0.1$ and noise amplitudes of 5%, 10%, 20%, 50% of the mean.

**Model architecture.** We used two types of models for this dataset, a black-box NeuralODE of Equation (1), and a gray-box NeuralODE used in Rackauckas et al. [54] that incorporates partial information about the underlying equation, given by:

$$\frac{dx}{dt} = \alpha x + U_1(x, y; \boldsymbol{\theta}_1), \quad \frac{dy}{dt} = -\gamma y + U_2(x, y; \boldsymbol{\theta}_2). \tag{10}$$

In our default setting, the black-box NeuralODE had 3 layers with [2, 32, 2] nodes respectively. The gray-box NeuralODE had 4 layers with [2, 20, 20, 2] nodes, with each node in the output layer corresponding to the output of $U_1$, $U_2$ of Equation (10). Following the results from [31], a non-saturating gelu activation was used for all layers except for the final layer, where identity activation was used. For the model capacity experiment (Figure 3, second panel), the number of nodes in the hidden layer was changed accordingly.

### E.2 Double pendulum

The double pendulum system is a canonical example in classical mechanics, and has four degrees of freedom $\theta_1, \theta_2, \omega_1, \omega_2$ corresponding to the angles and angular velocities of the two pendulums with respect to the vertical. The governing equation for the system can be derived using Lagrangian formulation of classical mechanics and is given by:

$$\frac{d\theta_1}{dt} = \omega_1, \quad \frac{d\theta_2}{dt} = \omega_2 \tag{11}$$

$$\frac{d\omega_1}{dt} = \frac{m_2 l_1 \omega_1^2 \sin \Delta\theta \cos \Delta\theta + m_2 l_2 \omega_2^2 \sin \Delta\theta + m_2 g \sin \theta_2 \cos \Delta\theta - (m_1 + m_2)g \sin \theta_1}{(m_1 + m_2)l_1 - m_2 l_1 \cos^2 \Delta\theta}, \tag{12}$$

$$\frac{d\omega_2}{dt} = -\frac{m_2 l_2 \omega_2^2 \sin \Delta\theta \cos \Delta\theta + (m_1 + m_2)(-l_1 \omega_1^2 \sin \Delta\theta + g \sin \theta_1 \cos \Delta\theta - g \sin \theta_2)}{(m_1 + m_2)l_2 - m_2 l_2 \cos^2 \Delta\theta} \tag{13}$$

where $\Delta\theta = \theta_2 - \theta_1$, $m_1, m_2$ are the masses of each rod, and $g$ is the gravitational acceleration.

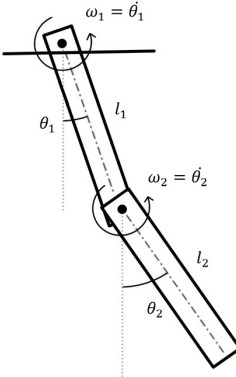

Figure 12: Diagram of a double pendulum.

**Data preparation.** While simulated trajectories for the double pendulum can be generated using the equations above, we instead used the experimental data from Schmidt & Lipson [59]. This consists of two trajectories from the double pendulum, captured using multiple cameras. The noise in the data is subdued due to the LOESS smoothing performed by the original authors. For our experiments, we used the first 100 points of the first trajectory for training and the next 100 to evaluate the extrapolation capabilities of the trained model.

**Model architecture.** Two types of models were used for this dataset: a black-box NeuralODE (Equation (1)) and a NeuralODE with second-order structure[23], which for this system, takes the form:

$$\frac{d\theta_i}{dt} = \omega_i, \quad \frac{d\omega_i}{dt} = U_i(\theta_1, \theta_2, \omega_1, \omega_2; \boldsymbol{\theta}_i); \quad i = 1, 2.$$

The black-box NeuralODE had 4 layers with [4, 50, 50, 4] nodes, with the input and output node numbers corresponding to the degrees of freedom of the system. The second-order model also had 4 layers with [4, 50, 50, 2] nodes. Note that there are now 2 output nodes instead of 4 since incorporating second-order structure requires the neural network to only model the derivatives of $\omega_1$ and $\omega_2$. For both models, gelu activations were used for all layers except the last, which used identity activation. Identical to the previous dataset, the NeuralODEs were integrated using the `dopri5` solver from the `torchdiffeq` package [11] with an absolute tolerance of 1e-9 and a relative tolerance of 1e-7.

### E.3 Lorenz system

The Lorenz system is given by the equations

$$\frac{dx}{dt} = \sigma(y - x), \quad \frac{dy}{dt} = x(\rho - z) - y, \quad \frac{dz}{dt} = xy - \beta z. \tag{14}$$

**Data preparation.** To generate the data, we followed experimental settings of Vyasarayani et al. [66] and used parameter values $\sigma, \rho, \beta = 10, 28, 8/3$ and the initial condition $x_0, y_0, z_0 = 1.2, 2.1, 1.7$. These conditions, upon integration, gives rise to the well-known "butterfly attractor". The training data was generated in the interval $t \in [0, 3.1]$, and still adhering to the paper, a Gaussian noise of mean 0 and standard deviation 0.25 was added to the simulated trajectories to emulate experimental noise. For data generation and NeuralODE prediction, the adaptive step size `dopri5` solver from the `torchdiffeq` package [11] was used with an absolute tolerance of 1e-9 and a relative tolerance of 1e-7. ODE solver and the tolerance values were kept identical to the previous Lotka-Volterra experiment.

**Model architecture.** We used a single NeuralODE for this dataset - a black-box model having 4 layers with [3, 50, 50, 3] nodes. The number of nodes for the input and output layers correspond to the three degrees of freedom of the state vector $\mathbf{u} = [x, y, z]^T$ of the system. The activations for the model was kept identical to the previous experiments - gelu activation for all layers, except for the last layer which used identity activation.

# F   Hyperparameter selection

For all combinations of models and datasets used, hyperparameters for the optimization were chosen by running sweeps prior to the experiment with a fixed random seed of 10, then selecting hyperparameter values that resulted in the lowest mean squared error value during training.

Note that for the experiment of Figure 3, the same hyperparameters used in Figure 4 was used for all experiments, due to the sheer cost of sweeping for the hyperparameters for every change of independent variables.

**Vanilla gradient descent.** Vanilla gradient descent has learning rate as its only hyperparameter. Due to finite computation budget, we set the maximum training epochs to 4000 for all sweeps and experiments. We list the values used in the hyperparameter sweep as well as the final selected values for each dataset in Table 1. We found that larger learning rates lead to numerical underflow in the adaptive ODE solver while training on the more difficult Lorenz and double datasets; hence, the sweep values for the learning rate parameters were taken to be lower than those for the Lotka-Volterra dataset.

| Dataset | Model | Hyperparameter | Sweep values | Selected value |
|---|---|---|---|---|
| Lotka-Volterra | Black-box 
 Gray-box | Learning rate | 0.005, 0.01, 0.02, 0.05, 0.1 | 0.05 
 0.02 |
| Double pendulum | Black-box 
 Second-order | Learning rate | 0.002, 0.005, 0.01, 0.02 | 0.02 
 0.05 |
| Lorenz | Black-box | Learning rate | 0.002, 0.005, 0.01, 0.02 | 0.005 |

Table 1: Hyperparameter sweep and selected values for vanilla gradient descent

**Multiple shooting.** The multiple shooting method has three hyperparameters: learning rate, number of segments, and continuity penalty. For the majority of experiments, we set number of segments to 5, and performed hyperparameter sweep on the remaining two parameters.

**Homotopy optimization.** Our homotopy-based NeuralODE training has five hyperparameters: learning rate ($\eta$), coupling strength ($k$), number of homotopy steps ($s$), epochs per homotopy steps ($n_{epoch}$), and homotopy parameter decrement ratio ($\kappa$).

While sweeping over all five hyperparameters would yield the best possible results, searching such a high dimensional space can pose a computation burden. In practice, we found that sweeping over only the first two hyperparameters and fixing the rest at predecided values still returned models that functioned much better than their vanilla counterparts. We list the predecided hyperparameter values for each dataset in Table 3. Note that we increased the epochs per homotopy step from 100 the Lotka-Volterra model to 300 for the Lorenz system and double pendulum datasets to account for the increased difficulty of the problem.

| Dataset | Model | Hyperparameter | Sweep values | Selected value |
|---------|-------|----------------|--------------|----------------|
| Lotka-Volterra | Black-box | Learning rate | 0.005, 0.01, 0.02, 0.05 | 0.05 |
| | | Continuity penality | 0.005, 0.002, 0.01 | 0.01 |
| | Gray-box | Learning rate | 0.005, 0.01, 0.02, 0.05 | 0.05 |
| | | Continuity penality | 0.005, 0.002, 0.01 | 0.002 |
| Double pendulum | Black-box | Learning rate | 0.005, 0.01, 0.02, 0.05 | 0.02 |
| | | Continuity penality | 0.005, 0.002, 0.01 | 0.002 |
| | Second-order | Learning rate | 0.005, 0.01, 0.02, 0.05 | 0.02 |
| | | Continuity penality | 0.005, 0.002, 0.01 | 0.002 |
| Lorenz | Black-box | Learning rate | 0.005, 0.01, 0.02, 0.05 | 0.02 |
| | | Continuity penality | 0.005, 0.002, 0.01 | 0.01 |

Table 2: Hyperparameter sweep and selected values for multiple shooting

| Dataset | Model | Hyperparameter | Predecided value |
|---------|-------|----------------|------------------|
| Lotka-Volterra | Black-box Gray-box | Number of homotopy steps | 6 |
| | | Epochs per step | 100 |
| | | $\lambda$ decay ratio | 0.6 |
| Double pendulum | Black-box | Number of homotopy steps | 7 |
| | | Epochs per step | 300 |
| | | $\lambda$ decay ratio | 0.6 |
| | Second-order | Number of homotopy steps | 6 |
| | | Epochs per step | 300 |
| | | $\lambda$ decay ratio | 0.6 |
| Lorenz | Black-box | Number of homotopy steps | 8 |
| | | Epochs per step | 300 |
| | | $\lambda$ decay ratio | 0.6 |

Table 3: Predecided hyperparameter values for homotopy optimization

Table 4 shows the sweeped hyperparameters as well as the final chosen values for each dataset. The total number of training epochs is given by multiplying the number of homotopy steps with epochs per homotopy steps. This resulted in 600 epochs for the Lotka-Volterra dataset, and 1800 epochs for both the Lorenz system and the double pendulum datasets.

| Dataset | Model | Hyperparameter | Sweep values | Selected value |
|---------|-------|----------------|--------------|----------------|
| Lotka-Volterra | Black-box | Learning rate | 0.01, 0.02, 0.05 | 0.05 |
| | | Control strength | 2, 3, 4, 5, 6 | 6 |
| | Gray-box | Learning rate | 0.01, 0.02, 0.05 | 0.02 |
| | | Control strength | 2, 3, 4, 5, 6 | 4 |
| Double pendulum | Black-box | Learning rate | 0.01, 0.02, 0.05 | 0.05 |
| | | Control strength | 3, 4, 5, 6, 7, 8, 9, 10 | 10 |
| | Second-order | Learning rate | 0.01, 0.02, 0.05 | 0.02 |
| | | Control strength | 3, 4, 5, 6, 7, 8, 9, 10 | 10 |
| Lorenz | Black-box | Learning rate | 0.005, 0.01, 0.02 | 0.02 |
| | | Control strength | 6, 7, 8, 9, 10 | 6 |

Table 4: Hyperparameter sweep and selected values for homotopy optimization

# G Further results

## G.1 Table of the benchmark results

Here, we present the benchmark results of Figure 4 in a table format as an alternative data representation.

Table 5: Experimental results for various NeuralODE models trained with three different methods. The values for best train epochs correspond to random seed values of 10, 20, and 30, except for runs marked with * where the backup random seed was used due to unstable training in the original random seed. MSE values are reported with the mean and standard error from three different runs.

| Dataset | Lotka-Volterra | | Double Pendulum | | Lorenz System |
|---|---|---|---|---|---|
| Model Type | Black Box | Gray Box | Black Box | Second Order | Black Box |
| Best train epochs | | | | | |
| Baseline | (3932, 3818, 3134) | (3999, 3980*, 3992) | (3998, 3731, 3633) | (2890, 3914, 3964) | (3960, 3996, 3958) |
| Multi-. Shoot. | (3644, 1482, 1359) | (3911, 2127, 1198) | (3438, 3454, 3646) | (3286, 2207, 3831) | (3710, 3666*, 3976) |
| Homotopy | **(299, 208, 227)** | **(291, 265, 216*)** | **(1201, 2087, 1502)** | **(1200, 600, 900)** | **(2399, 2399, 2377*)** |
| Mean Squared Error ($\times 10^{-2}$) | | | | | |
| Baseline (interp.) | 1.76±0.13 | 27.3±8.68 | 1.61±0.12 | 1.03±0.12 | 259±6.05 |
| Multi-. Shoot.(interp.) | 0.95±0.02 | 0.74±0.02 | **0.81±0.23** | 0.28±0.03 | **12.9±1.03** |
| Homotopy (interp.) | **0.78±0.05** | **0.72±0.01** | 0.98±0.08 | **0.18±0.02** | 18.6±1.75 |
| Baseline (extrap.) | 2.73±0.17 | 6254±5026 | 944.9±159.7 | 1619±172.9 | 603±1.36 |
| Multi-. Shoot.(extrap.) | 1.42±0.01 | 262.2±212.4 | 2676±391.7 | 2365±105.1 | 138±37.0 |
| Homotopy (extrap.) | **1.22±0.09** | **7.656±4.557** | **1031±103.2** | **428.5±51.90** | **92.8±2.59** |

## G.2 Example of the training curves

Here, we include some of the training curves for our experiments. Figure 13 displays the averaged training curves for the black-box model trained on the Lotka-Volterra dataset. From the right panel, it can clearly be seen that our homotopy method optimizes the MSE much rapidly than the other methods, arriving at the noise floor in less than 500 epochs. The abrupt jumps in the MSE curve for homotopy is due to the discontinous change in the train loss that occurs every the the homotopy parameter is adjusted. To make this connection clearer, we show the train loss as well as the homotopy parameter for our method in the left panel of the same figure.

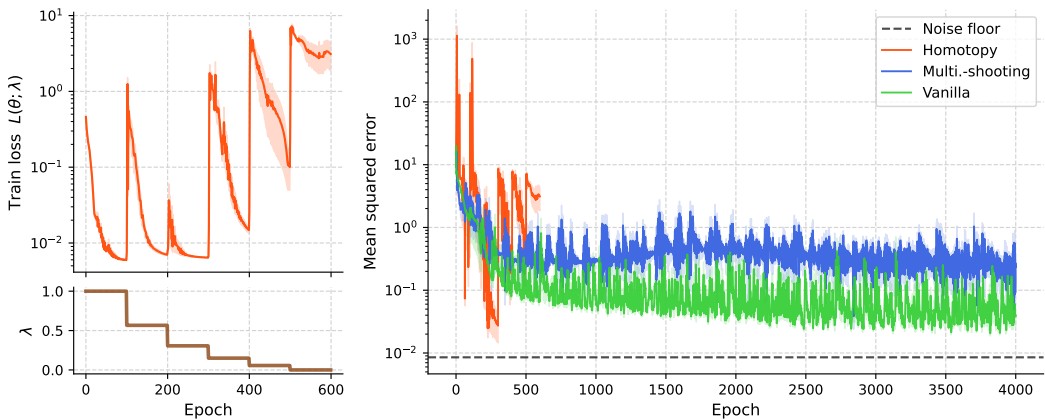

Figure 13: Training curves for the black-box model on the Lotka-Volterra dataset, corresponding to the benchmark results of Figure 4. **(Left)** Train loss $\mathcal{L}(\boldsymbol{\theta}, \lambda)$ and the homotopy parameter $\lambda$. **(Right)** Mean squared error as a function of training epochs. Note that the curves for homotopy stops early due to the choice of the algorithm hyperparameters (see Appendix F for further details).

We present analogous results for the second-order model trained on the double pendulum dataset in Figure 14. Once again, we find that our homotopy method converges much quicker than its competitors.

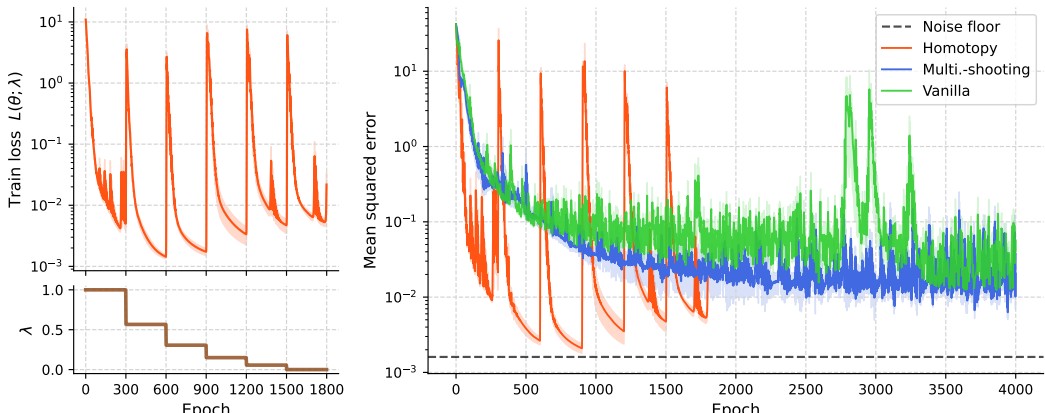

Figure 14: Train loss, homotopy parameter, and mean squared error as a function of epochs for the results of Figures 4, 5 and 6. The layout of the plots, as well as their interpretations are identical to that of Figure 13.

## G.3  Further discussion on the Lotka-Volterra system results

In this section, we present additional results regarding our experiment of Figure 3 and continue our discussion.

**Increasing data length.**  Figure 15 depicts the predicted trajectories for differently trained models. For the shortest data, it can be seen that all models did overfit on the training data and failed to capture the periodic nature of the system. This justifies our attributing the large extrapolation error of the models in Figure 3 to overfitting. However, we emphasize that this overfitting is not due to failings in the training algorithms, but rather due to the insufficient information in the training data, as the models cannot be expected to learn periodicity without being given at least a single period worth of data. As the data is increased, we find that both homotopy and multiple shooting properly capture the dynamics of the system, whereas the vanilla method was unable to properly learn the system for the longest data.

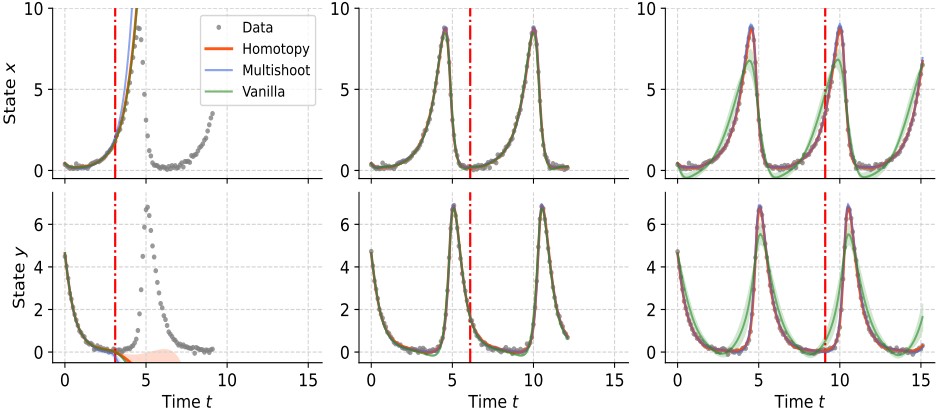

Figure 15: Predicted trajectories for the models with increasing train data length, t=(3.1, 6.1, 9.1). Corresponds to the first panel of Figure 3. The red dashed line indicates the start of extrapolation.

**Reducing model capacity.**  In Figure 16, we show the prediction trajectories corresponding to the model capacity experiment. It is clear from the results that both the homotopy and multiple shooting methods can produce models that accurately portray the dynamics, regardless of the reduction in the model size. In contrast, predictions from vanilla training deteriorate as the model size decreases, with the smallest model inaccurately estimating both the amplitude and the phase of the oscillations.

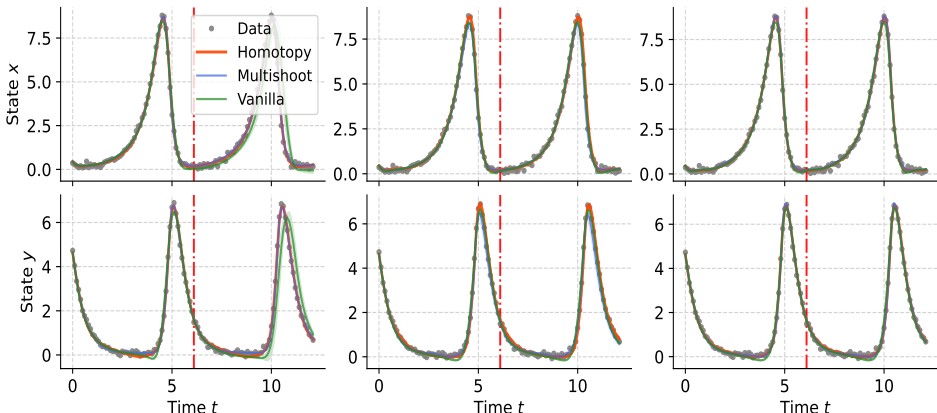

Figure 16: Training curves for each number of nodes, nodes=(8, 16, 32). Corresponds to the second panel of Figure 3.

**Increasing sampling period.**    The model trajectories for decreasing sampling period in the training data is shown below in Figure 17. Intriguingly, we find that the multiple shooting method results suffer greatly with the increased data sparsity. This is a side effect of our experiment setting, where we set the time interval constant while increasing the sampling period. To elaborate, our choice of fixed time interval causes the number of training data points decrease as the period increases. However, as multiple shooting trains by subdividing the training data, it struggles on small datasets because this leads to each segment containing even less data points that do not convey much information about the data dynamics.

Another interesting observation that can be made is that vanilla training gives better results when sparser data is used, which is also confirmed by the error values in the third panel of Figure 3. This suggests an alternate training improvement strategy for training NeuralODEs on long time series data - by training on subsampled data, then if necessary, gradually anneal the data intervals back to its original form as training proceeds. Of course, the effectiveness of such a scheme will need to be tested for more complex systems, which is outside the scope of this paper.

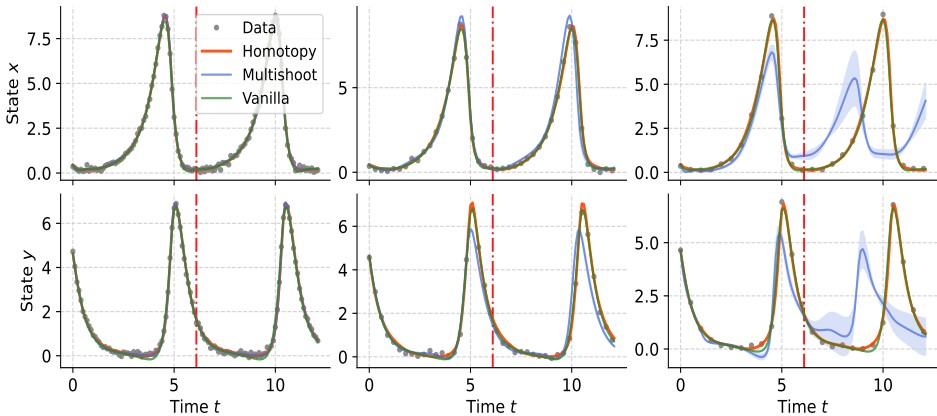

Figure 17: Predicted trajectories for the models with increasing sampling periods, dt = (0.1, 0.3, 0.5). Corresponds to the third panel of Figure 3.

As our homotopy training method utilized a cubic smoothing spline to supply the coupling term, we also inspected the quality of this interpolant as the data period was increased. From Figure 18, we find that the cubic spline relatively resistant to the increased data sparsity, which may be one of the reasons why our homotopy method is very robust against this mode of data degradation.

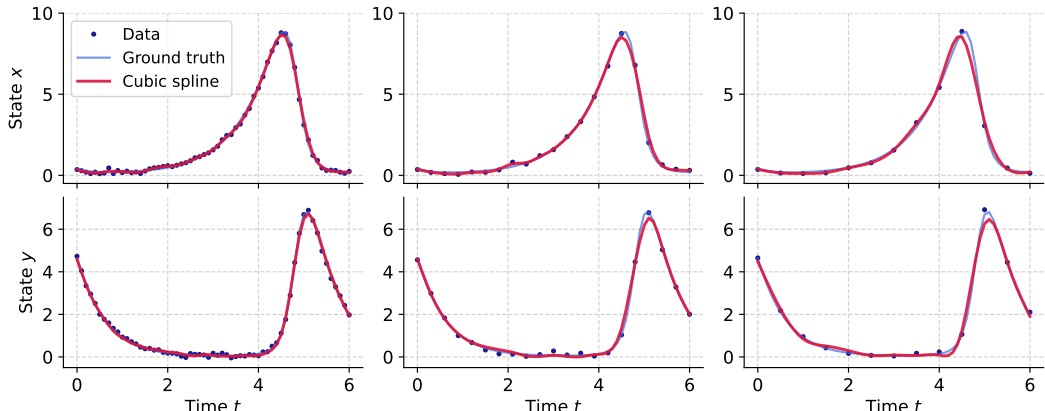

Figure 18: Cubic spline interpolation results for data with increasing sampling periods, dt=(0.1, 0.3, 0.5). Corresponds to the third panel of Figure 3.

**Increasing data noise.** Finally, Figure 19 shows the model trajectories as the noise in the training data is increased. Here, we do find that while all methods return deteriorating predictions as the noise is increased, our homotopy method is found to be most robust to noise, followed by multiple shooting, and finally vanilla gradient descent. A part of our algorithm's robustness to noise can be explained by the use of the cubic smoothing splines. As our use of synchronization couples the NeuralODE dynamics to the cubic spline trajectory during training, one could argue that the denoising effect of cubic splines is thus directly transferred to the model.

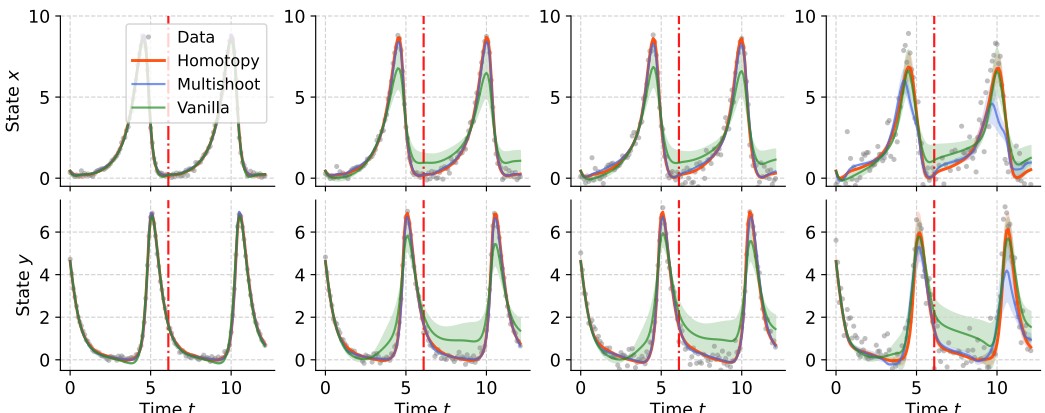

Figure 19: Predicted trajectories for the models with increasing noise, noise=(0.05, 0.1, 0.2, 0.5). Corresponds to the fourth panel of Figure 3.

However, this does not seem to be the full picture. Figure 20 shows the cubic smoothing splines used during modeling training for each of the noise levels. As we held the amount of smoothing constant throughout our experiments, we see that for larger noise amplitudes, the spline fails to reject all noise, and displays irregular high frequency behaviors. On the other hand, the corresponding trained trajectories for the homotopy method (Figure 19, third and fourth panels) do not mirror such irregular oscillations, indicating that the homotopy optimization procedure itself also has an intrinsic robustness to noise.

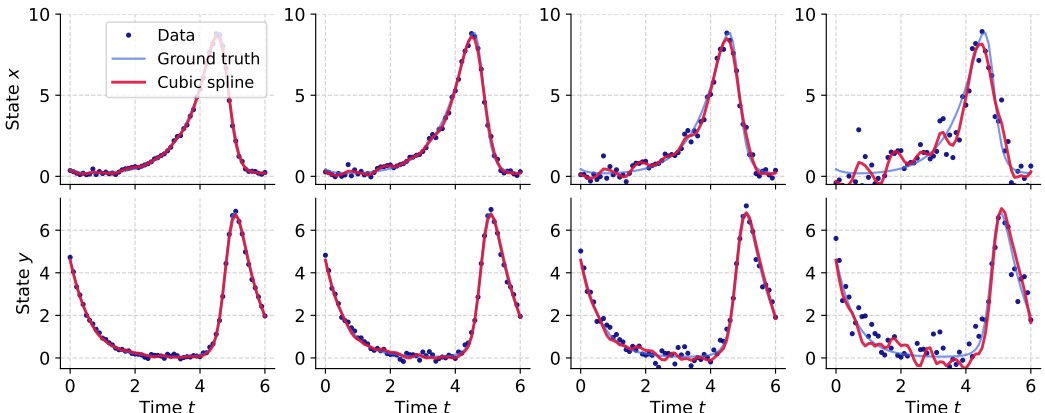

Figure 20: Cubic spline interpolation results for data with increasing noise amplitude, noise=(0.05, 0.1, 0.2, 0.5). Corresponds to the fourth panel of Figure 3.

### G.4 Comparison to a non-deep learning algorithm: SINDy

In this section, we briefly compare our algorithm to a more traditional symbolic regression-based method. We choose the well-known SINDy algorithm, which is readily available in the `pysindy` package. This algorithm takes as input, time series measurements of the states of the dynamical system in question, as well as a dictionary of candidate terms that could constitute the governing equation. Afterwards, the time derivative of the states is estimated numerically, and a sparse regression is performed to determine which terms exist in the governing equation, as well as what their coefficient values are.

Compared to neural network-based approaches, the unique feature of SINDy is that its results are given in the form of analytical expressions. Therefore if the governing equation for the data is simple (e.g. consists of low-order polynomial terms), or if the user has sufficient prior information about the system and can construct a very small dictionary that contains all the terms that appear in the true equation, SINDy can accurately recover the ground truth equation.

On the other hand, if the system is high dimensional so that the space of possible candidate terms start to grow drastically, or if the governing equation has a arbitrary, complex form that cannot be easily guessed, SINDy either does not fit the data properly, or even if it does, recovers inaccurate equations that are difficult to assign meaning to. These correspond to situations where NeuralODEs, and hence our method for effectively training them, shines.

To illustrate this point, we choose the double pendulum dataset and compare the predicted trajectories from SINDy and our homotopy algorithm. As one can see from Equation (13), the governing equation of this system has an extremely complicated form that cannot be guessed easily. To reflect this difficulty, we chose the basis set of the dictionary of the candidate terms to be the state variable themselves, their sines and cosines, and the higher harmonics of the sines and cosines.

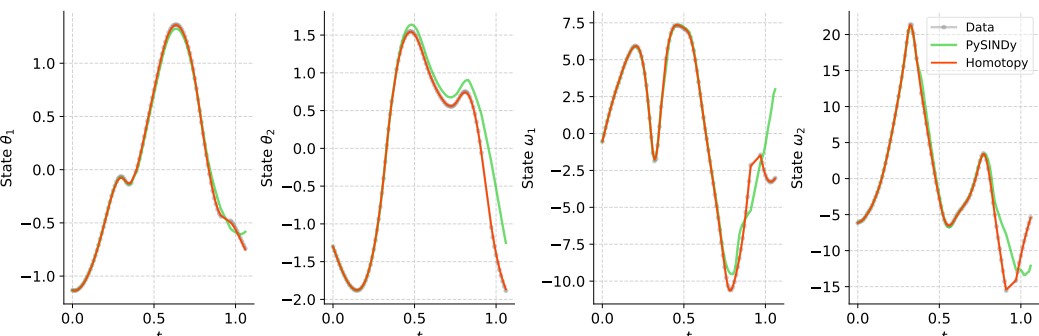

Figure 21: Comparison between our homotopy method and SINDy on the double pendulum dataset.

From Figure 21, we find that SINDy is able to fit the data to a modest accuracy. While this may seem weak compared to the result from our homotopy method, the accuracy of the SINDy prediction is likely to improve if the candidate dictionary is updated to better reflect the actual form of Equation (13). However, this amounts to supplying the algorithm with excessive amounts of prior information about the system, which is rarely available in practice. This is also constrasted with NeuralODE training with our homotopy method - which does not require any additional information about the system to produce the results above.

### G.5 Changing the gradient calculation scheme

As we commented in Section 3, methods that focus on other aspects of NeuralODE training, such as alternate gradient calculation schemes, are fully compatible with our homotopy training method. To demonstrate this point, we used the symplectic-adjoint method [39] provided in the `torch-symplectic-adjoint` library as a substitute for the direct backpropagation used for gradient calculation in our experiments.

Table 6: Benchmark results with the gradient calculation scheme changed to the symplectic-adjoint method.

| Dataset | Lotka-Volterra | | Double Pendulum | | Lorenz System |
|---|---|---|---|---|---|
| Model Type | Black Box | Gray Box | Black Box | Second Order | Black Box |
| Best train epochs | | | | | |
| Baseline | (3999, 3999, 3853) | (3997, 3999, 3992) | (3909, 3384, 3998) | (3983, 3998, 3972) | (3994, 3999, 3990) |
| Multi-. Shoot. | (3982, 3968, 3670) | (3901, 3958, 3996) | (3805, 3913, 3820) | (3988, 3970, 3942) | (3977, 3418, 3954) |
| Homotopy | **(299, 208, 228)** | **(598, 265, 291)** | **(1201, 1799, 1774)** | **(900, 1799, 900)** | **(1799, 918, 1799)** |
| Mean Squared Error ($\times 10^{-2}$) | | | | | |
| Baseline (interp.) | 2.47±0.13 | 187±133 | 1.69±0.20 | 1.02±0.12 | 184±60.1 |
| Multi-. Shoot.(interp.) | 1.20±0.02 | **0.95±0.02** | **0.76±0.01** | 0.61±0.02 | **29.1±15.8** |
| Homotopy (interp.) | **0.77±0.04** | 1.68±0.76 | 1.08±0.12 | **0.25±0.03** | 33.3±5.43 |

Comparing the above results to our benchmark results in Figure 4 and Table 5, we see that the overall results remain the same - our homotopy method is able to train the models effectively, in much smaller number of epochs.

