# OpenReview forum: "Homotopy-based training of NeuralODEs for accurate dynamics discovery"
_NeurIPS.cc/2023/Conference — NeurIPS 2023 poster_

### Official Review · Reviewer_U4Km · 2023-07-04

**Soundness:** 3 good
**Presentation:** 3 good
**Contribution:** 2 fair
**Rating:** 6
**Confidence:** 4

**Summary:**

This paper proposes a novel method for training NeuralODEs, based on synchronization and homotopy optimization. They show that the addition of the synchronization module can smooth the loss landscape, on which homotopy optimization can be applied to enhance training. The new training method achieves competitive results in convergence speed and interpolation and extrapolation accuracy when compared with other baseline methods, especially for long training data. In addition,they demonstrate the robustness of the method experimentally.


**Strengths:**

1. The method of homotopy optimization has been introduced into the training of various neural networks, but this paper combines the idea of homotopy optimization with synchronization, and proposes a method that is very suitable for training NeuralODE on dynamical system-related tasks.
2. The new training method can effectively improve the efficiency of network training and has the potential to alleviate a series of problems caused by irregular loss landscape.
3. Compared with other methods, this method has no restrictions on the model structure or parameters at all, so it is more general.
4. The experiments are valid.

**Weaknesses:**

1. The explanations in this paper are all experimental and lack rigorous theoretical support. For example, whether it is possible to analyze if the hessian trace of loss can be strictly bounded by the traditional method after adding coupling items for homotopy optimization.
2. The authors only test the proposed method in low-dimensional toy models.
3. It is mentioned in the paper that there are many stabilized models that improve training by limiting the expressivity of the model, but these methods are not compared experimentally with this new method to verify the advantages of this new method.


**Questions:**

1. Is it possible to add some theoretical discussions on the smoothness of the loss landscape？
2. Could you provide experimental results on more highly complex real-world datasets?


**Limitations:**

The authors have adequately addressed the limitations.

---

> ### Author Rebuttal · Authors · 2023-08-10
>
> We thank the reviewer for the valuable comments and the constructive feedback. We have added the responses to both of the reviewer's questions in the global response above. We look forward to additional discussions with the reviewer.
>
> We also add a bit more theoretical description about the synchronization below:
>
> Let's say that our system under consideration is $$d\mathbf{x}(t)/dt=\mathbf{F}(\mathbf{x}(t)),$$
> where $\mathbf{x} \in \mathbb{R}^m$ is an m-dimensional vector. We take the input vector to be transformed as a scalar variable in the form $u(t)=\mathbf{K}^T\mathbf{x}(t)=K_1x_1(t)+K_2x_2(t)+...+K_mx_m(t)$, with $\mathbf{K}$ a constant column vector.
>
> The output subsystem is then written as $$d\mathbf{y}(t)/dt=\mathbf{F}(\mathbf{y}(t))-\mathbf{D}(v(t)-u(t))$$, where $v(t)=\mathbf{K}^T\mathbf{y}(t)$ and $\mathbf{D}=(D_1, D_2, ..., D_m)^T$ is an m-dimensional constant vector.
>
> Observe that if $\mathbf{y}(t)=\mathbf{x}(t)$ is plugged into Eq. (2), the equation is satisfied, meaning that synchronization of the systems is possible for the combined system, Eqs. (1) and (2). It can be guaranteed that if we evaluate the largest Lyapunov exponent for the subsystem Eq. (2) with respect to the trajectory $\mathbf{y}(t)=\mathbf{x}(t)$.
>
> Consider infinitesimal deviations of $\mathbf{y}(t)\$ from $\mathbf{x}(t)$,
> $$
> \mathbf{y}(t)=\mathbf{x}(t)+\delta\mathbf{y}(t).
> $$
> From (2),
> $$
> d\delta\mathbf{y}(t)/dt=[\partial\mathbf{F}(\mathbf{y})/\partial\mathbf{y}\mid_{\mathbf{y}=\mathbf{x}}-\mathbf{DK}^T]\delta\mathbf{y}.
> $$
>
> Here, if you see the $[\partial\mathbf{F}(\mathbf{y})/\partial\mathbf{y}\mid_{\mathbf{y}=\mathbf{x}}-\mathbf{DK}^T]$ as a matrix A and for $\delta\mathbf{y}$ as an error vector, then the Eq.(3) can be rewritten as:
>
> \begin{equation}
> d\mathbf{e}/dt=\mathbf{A}\mathbf{e}
> \end{equation
>
> The solution of this differential equation is given by
>
> \begin{equation}
> \mathbf{e}(t)=e^{\mathbf{A} t}\mathbf{e}(0)
> \end{equation}
>
> where $\mathbf{e}(0)$ is the initial value of $\mathbf{e}(t)$.
>
> We can always make this to be converged by choosing the proper value of $\mathbf{K}$.
>
> Connecting this result to the hessian of the loss is a bit tricky at the moment as the matrix $\bf A$ depends on the parameters as well, and we are continuing to research on this front at the moment.
>
> We suspect it might be the case that it is impossible to state a global smoothness claim for the loss landscape. However, we believe by drawing on the discussion of homotopy from topology, it can be possible to weakly assert the local smoothness of the loss manifold.
>
> The main general idea in topology is to study spaces which can be continuously deformed into one another. This idea is given mathematical substance by the introduction of homeomorphisms.
>
> If we take two topological spaces $T_1$ and $T_2$ then a map $\alpha$ from $T_1$ to $T_2$:
>
> \begin{equation}
> \alpha : T_1 \rightarrow T_2.
> \end{equation}
>
> is called a homeomorphism if it is both continuous and has an inverse which is also continuous. This notion, homotopy, is inspired, as is the notion of homeomorphism, by the more informal or intuitive notion of deformation.
>
> While homeomorphism generates equivalence classes whose members are topological spaces, homotopy generates equivalence classes whose members are continuous maps. Take two continuous maps $\alpha_1$ and $\alpha_2$ from a space $X$ to a space $Y$:
>
> \begin{equation}
> \begin{split}
> \alpha_1 : X \rightarrow Y, \\
> \alpha_2 : X \rightarrow Y.
> \end{split}
> \end{equation}
>
> Then the map $\alpha_1$ is aid to be homotopic to the map $\alpha_2$ if $\alpha_1$ can be deformed into $\alpha_2$, in precise mathematical terms:
>
> \begin{equation}
> F:X \times [0, 1] \rightarrow Y,\quad F \ continuous.
> \end{equation}
>
> and $F$ satisfies
>
> \begin{equation}
> \begin{split}
> F(x, 0)=\alpha_1(x), \\\\
> F(x, 1)=\alpha_2(x).
> \end{split}
> \end{equation}
>
> In other words as the real variable $t$ in $F(x, t)$ varies continuously from 0 to 1 in the unit interval $[0, 1]$; the map $\alpha_1$ is deformed continuously into the map $\alpha_2$. Homotopy is an equivalence relation and it divides the space of continuous maps from $X$ to $Y$, which is written $C(X, Y)$, into equivalence classes. Since a homeomorphism is also a continuous map, these equivalence classes are unchanged under homeomorphism of $X$ or $Y$.

---

> > ### Comment · Reviewer_U4Km · 2023-08-12
> >
> > Thank you for your reply. Although no strict theoretical proof has been given, the above explanation explains the rationality of the method to a certain extent. However, its extrapolation prediction on more complex real-world systems is poor, which shows that its applicable scenarios are very limited. For the above reasons, I maintain the previous score.

---

### Official Review · Reviewer_84Va · 2023-07-05

**Soundness:** 3 good
**Presentation:** 4 excellent
**Contribution:** 3 good
**Rating:** 7
**Confidence:** 4

**Summary:**

Training neural ODE models on long sequences of data from a dynamical system is difficult. The authors argue empirically that this is due to a poorly conditioned loss landscape leading to difficulties in optimization. To rectify this difficulty the authors use tools from the literature on synchronization (which gives conditions for when two dynamical systems will converge based on an error correction coupling term. They provide a homotopy between the synchronized dynamical system and the original one and argue that the optimization dynamics will be nice on the sychronized system and progressively less nice along the homotopy to the original system. However by gradually training the parameters and moving along the homotopy the method “ratchets up the difficulty” in a way that makes the ultimate optimization problem easier.

They demonstrate the success of the system on a collection of commonly used dynamical systems taken from the neural ODE literature when compared to vanilla SGD and a multiple-shooting algorithm that handles the longer time horizons by breaking them up. The authors show that their method archives better accuracy, robustness, and convergence than the baselines on these problems.

**Strengths:**

The work combines the idea of synchonization with the standard Neural ODE training to improve the performance on long-horizon data. Since “long-horizon” data is not that long when training Neural ODEs, this is a welcome development. The idea of synchronization is intuitive and theoretically justified as well as empirically demonstrated to be useful through the loss landscape Hessian. The empirical results are not broad but they are deep; the authors study many variations of the problem on a small number of dynamical systems (3) against a small number of baselines (2). However, the method performs well under noise, sparsity, and long data on these and with better test performance and more quickly.


**Weaknesses:**

The theoretical explanation leaves a little bit lacking. I would love to know more about the properties of the dynamical system as it is more or less synchronized. If there were traces of such in an image I think it would be intuitive. It took me a while to work out that as \lambda -> \infty any setting of the parameters is optimal and I think it might be good to explicitly say / visualize that. Any more detailed theoretical statements about this method would also be welcomed.


**Questions:**

How is K chosen? I couldn’t find it in the main paper and the idea is intuitive even if K=I but I’d like to know?

Would this method work for control systems as well? Seems like the counterfactual nature of the problem could be interesting.


**Limitations:**

This seems pretty general and seems to widely apply to the breadth of neural ODE architectures and dynamical systems.

---

> ### Author Rebuttal · Authors · 2023-08-10
>
> We thank the reviewer for the interest in our paper and the positive comments. We have listed the answers to the reviewer's question below, and have grouped the answers to some of the more commonly occurred questions from multiple reviewers in the global response above.
>
> > The theoretical explanation leaves a little bit lacking. I would love to know more about the properties of the dynamical system as it is more or less synchronized. If there were traces of such in an image I think it would be intuitive.
>
> We had originally tried to illustrate the effect of coupling strength on the dynamics trajectories through Figure 10 in Section B of the Appendix  (please look at the appendix.pdf file in the zip file of our Supplementary materials as it is the updated version compared to the one at the end of the main paper pdf). In the figure, we perturb the coefficients of the Lotka-Volterra and Lorenz systems, couple them to a reference trajectory, then plot the resulting dynamics trajectory as well as the loss landscape as a function of coupling strength. We hope this can provide some additional information about our work.
>
> We would also love to add additional visualizations the reviewer finds would be useful and look forward to the reviewer’s comments.
>
> > It took me a while to work out that as $\lambda \rightarrow \infty$ any setting of the parameters is optimal and I think it might be good to explicitly say / visualize that.
>
> In our paper, we wanted to convey this point through the upper right panel of Figure 2, where we drew the loss landscape for increasing values of the coupling strength. We will update this part to explicitly mention that the flat landscape for the large value of k means that any setting of the parameters is equally optimal. Also to better visualize the flatness of the landscape, we will adjust the y-axis limits as well.
>
>
> > How is K chosen? I couldn’t find it in the main paper and the idea is intuitive even if K=I but I’d like to know?
> We apologize for the confusion. In our work, the matrix $\bf K$ and the scalar algorithm hyperparameter $k$ are related to each other by ${\bf K}=k\bf I$, where $\bf I$ is the identity matrix.
>
> We would like to expound on this by clarifying that this is not the only way to construct the coupling matrix – any choice of $\bf K$ can achieve synchronization provided that the matrix $\frac{\partial \bf U}{\partial \bf u}-\lambda\bf K$ is always negative definite along the trajectory ${\bf u}(t)$. However, naively constructing $\bf K$ in a different manner results in additional hyperparameters corresponding to each of the independent elements of $\bf K$. Therefore, we decided to reduce $\bf K$ into a single scalar value to minimize the number of hyperparameters in our algorithm.
>
> With that being said, we do think that exploring different parametrization of $\bf K$ is an important future direction of our research. In systems where the order of magnitude of the different degrees of freedom differ significantly, it can be more intuitive to use ${\bf K}=diag(k_1, k_2 , \dots)​$.  Also in the chaos synchronization literature, [1] reports that $K = BV^T$ where $B, V$ are constant vectors can be tuned to achieve synchronization when the parameters of the coupled systems differ greatly.
>
> > Would this method work for control systems as well? Seems like the counterfactual nature of the problem could be interesting.
>
> We thank the reviewer for the very interesting discussion point. We believe that our method can work on control systems, but the added complexity of the problem will require additional experimentation and theoretical studies to accurately confirm our hypothesis.
>
> To elaborate, applying our homotopy method to control system is straightforward since everything can be kept the same and only the NeuralODE slightly modified to accept the control signal ${\bf v}(t)$:
>
> $$
> \frac{d\bf u}{dt}={\bf U}(t,\bf u, v;\theta)-K(u-\hat u)
> $$
>
> where $\bf \hat u$ is the interpolant constructed from measured time series data ${\bf u}_{data}(t_i)$.
>
> As for whether this scheme will actually result in improved training, we can take some hints from the synchronization literature. In [2], synchronization and homotopy optimization were applied to estimate the coefficients of ODE systems that contain time-dependent forcing terms. Since the such forcing terms can be viewed as control terms, this study shows that it is indeed possible to use the homotopy method to discover control systems and outline the possibility of using our homotopy method to train NeuralODEs on such systems as well.
>
> [1] G. A. Johnson et. al, *Phys. Rev. Lett.* **80**, 18 (1998).
>
> [2] R. Manikantan et. al, *Mathematics* **8**, 7 (2020).

---

### Official Review · Reviewer_bK9U · 2023-07-06

**Soundness:** 3 good
**Presentation:** 3 good
**Contribution:** 3 good
**Rating:** 6
**Confidence:** 3

**Summary:**

The authors present a new method for training Neural Ordinary Differential Equations (NeuralODEs), a modeling approach that merges neural networks with the paradigm of differential equations from physical sciences. While NeuralODEs offer significant potential for extracting dynamic laws from time series data, they often suffer from long training times and subpar results. In this paper, the authors propose a training method built on synchronization and homotopy optimization. Their results demonstrate that the new method achieves competitive or even better training loss, often requiring fewer epochs compared to traditional model-agnostic techniques, and also exhibits better extrapolation capabilities.

**Strengths:**

1. The paper addresses a notable challenge in the field of NeuralODEs: the long training times and subpar results. Their proposed solution is both innovative and logical, introducing a novel concept in the NeuralODE literature.
2. The authors demonstrate the advantages of their method convincingly, showing that it improves the training loss and requires fewer epochs for training.

**Weaknesses:**

1. The method's extension to high-dimensional systems or partially observed states is yet to be explored, limiting its current applicability.
2. The paper could have benefited from more real-world application examples, demonstrating the utility of the method beyond the context of benchmark experiments.

**Questions:**

1. How does the efficacy of homotopy and multiple-shooting methods change with the size of the NeuralODE model? Is it possible to find a balance between model size and training method for optimal performance?
2. What are the effects of data sparsity and noise on the performance of different training methods for NeuralODEs, particularly the homotopy training method?
3. How can this methodology be extended to high-dimensional systems or systems with partially observed states?
4. What are the practical implications and potential applications of this training method in various industries or scientific disciplines?

**Limitations:**

The paper mainly uses synthetic datasets for testing the performance of training algorithms. The findings might not generalize to real-world datasets, which can exhibit unique complexities and noise structures.

---

> ### Author Rebuttal · Authors · 2023-08-10
>
> We thank the reviewer for the comments as well as the detailed questions. We have listed the responses to the questions below and grouped the responses to some of the more common questions in the global comment above.
>
> > How does the efficacy of homotopy and multiple-shooting methods change with the size of the NeuralODE model? Is it possible to find a balance between model size and training method for optimal performance?
>
> While more experiments comparing the two methods are needed for a final confirmation, we found in our experiments that the relative efficacy of the two methods were not very sensitive to the model size. Both methods returned orders of magnitude improvements compared to the vanilla training albeit homotopy method resulting in better extrapolation performance than the multiple-shooting method.
>
> However, continuing the discussion, we do think there is a different factor that can change the relative efficacy of the two methods -  the length of the training data. This is because the multiple-shooting method can be implemented to compute all of the shooting segments parallel in time [1] - resulting in a time complexity proportional to the length of one segment - whereas the homotopy method scales with the length of the total data.
>
> Therefore, for short to long time series data, homotopy training method tends to be more effective than multiple shooting because even though a single epoch of the homotopy method can take longer than multiple shooting, its faster convergence and improved extrapolation qualities makes homotopy the more attractive option. For very long sequences however, multiple-shooting method can be more economical as the reduced time complexity starts to outshine the improved convergence speed of the homotopy method. This is a very interesting direction of research that we would like to pursue in the future.
>
> Another potentially interesting direction is to use the homotopy method and the multiple-shooting method at the same time - that is, subdivide the data into segments and apply synchronization on each - as the two frameworks are compatible with each other. This has the potential to reap the benefits of the two methods at the cost of increased number of parameters (the sum of the hyperparameters of the two methods).
>
> > What are the effects of data sparsity and noise on the performance of different training methods for NeuralODEs, particularly the homotopy training method?
>
> Thank you for your question. Regarding the performances of the different training methods with respect to data sparsity and noise, we would like to refer the reviewer to the last paragraph of Section 7.1 (corresponding to third and fourth columns of Figure 3) and Section G.3 in the appendix with emphasis on Figures 17-20 (please look at the appendix.pdf file in the zip file of our Supplementary materials as it is the updated version compared to the one at the end of the main paper pdf).
>
> First, regarding noise, we find that the vanilla method is very sensitive with the model predictions degrading rapidly with increasing noise. In comparison, both the homotopy and the multiple-shooting methods are much more robust, although for very large noise amplitudes our homotopy method prevails. An interesting point here is that large amounts of noise can introduce high frequency artifacts to the cubic spline interpolant used in homotopy training (Figure 20, rightmost column) but the corresponding training results (Figure 19, rightmost column) are not affected by these artifacts.
>
> As for data sparsity, we find that multiple-shooting method suffers the most as data sparsity increases. This is intuitive because as multiple-shooting method subdivides the time series data, it works well as large, dense data regime where each shooting segment is given enough training data elucidate the system dynamics. Another unexpected phenomenon we observed that the vanilla method tends to work better as the data sparsity increases. This hints at another potential training improvement strategy where one first trains a NeuralODE with an undersampled version of the data, then subsequently training the model on denser and denser version of the data to arrive at the final training result. Finally, we find that our homotopy method is quite robust to data sparsity - model predictions do degrade as data sparsity is increased, but still the training model predictions are competitive with those from vanilla training. As the homotopy method produces such results at an accelerated training time, we believe homotopy method is still better in this situation.
>
>
> > What are the practical implications and potential applications of this training method in various industries or scientific disciplines?
>
> Our homotopy training method is valuable in multiple scientific disciplines where the goal is to discover governing equations underlying data or need to create a surrogate dynamics model that provides reliable forecasts of the data.
>
> One example is [2], where the authors introduced a gray-box model to predict the temperature of the ocean as a function of depth. The neural network component in this hybrid model serves to account for the missing physics that are not captured by traditional physics based modeling.
>
> To obtain a proper surrogate model, an effective training method is necessary and this is where our homotopy method shines. Especially, our results on the gray-box models for the Lotka-Volterra system (main paper, Figure 4) show that even though significant prior knowledge on the governing equation was added to the NeuralODE model, vanilla training was not able to fully capitalize on this and instead returned models the performed significantly worse than their black-box counterparts. Our homotopy method can help resolve this problem, thus aiding in creating proper surrogate models.
>
> [1] M. Stefano et. al, Differentiable Multiple Shooting Layers. *NeurIPS* **34** (2021).
>
> [2] A. Ramadhan et. al. *arXiv:2010.12559* (2023).

---

> > ### Comment · Reviewer_bK9U · 2023-08-12
> >
> > The author has substantially addressed most of my concerns. I will keep my score.

---

### Official Review · Reviewer_jYQA · 2023-07-07

**Soundness:** 3 good
**Presentation:** 4 excellent
**Contribution:** 3 good
**Rating:** 6
**Confidence:** 4

**Summary:**

The paper proposes a homotopy-based training method for NeuralODE models, in particular for the case with cases with long sequence of training data.  Comprehensive experimental results demonstrated the effectiveness of the proposed method.

**Strengths:**

1. The proposed method is based on homotopy method, which is a classic global optimization method, with sound theoretical foundation.
2. Comprehensive experimental results. Good results when compared with the baseline methods
3. The paper explains key concepts and results well.

**Weaknesses:**

1.  The argument on the loss function landscape lacks theoretical insights. The connection is established by empirical results.
2.  Although the overall idea of using homotopy is good, some details are not properly addressed. See the questions and limitation part for more details.
3. A few key references related to homotopy methods are not cited. The references by JH He really put homotopy methods in the mainstream of scientific computing: https://scholar.google.com/citations?user=tzM7c2cAAAAJ&hl=en


**Questions:**

1. I question the statement around Line 196, on the usage of cubic splines and its potential consequences. Cubic spline is a relatively high-order method that may be inaccurate when the underlying dynamic system is **stiff** (or there are widely spread time constants). Will the cubic spline interpolation be able to handle responses from _stiff_ systems?

2. I question the convergence of the homotopy method when constant $\Delta \lambda$ is used. The introduction of homotopy to the original ODE effectively increases the order of the ODE by 1, and transform it into a DAE.  The authors should address whether there is the potential of non-convergence, and when it happens, how to address it.

**Limitations:**

No negative societal impact perceived.

---

> ### Author Rebuttal · Authors · 2023-08-10
>
> We thank the reviewer for the comments and the very interesting questions. We list our answers below, and have grouped the common questions in the global comment.
>
> > A few key references related to homotopy methods are not cited. The references by JH He really put homotopy methods in the mainstream of scientific computing: https://scholar.google.com/citations?user=tzM7c2cAAAAJ&hl=en
>
> Thank you for kindly pointing us to key references regarding the homotopy method. We will update the manuscript to include these references in the related works section.
>
> > I question the statement around Line 196, on the usage of cubic splines and its potential consequences. Cubic spline is a relatively high-order method that may be inaccurate when the underlying dynamic system is **stiff** (or there are widely spread time constants). Will the cubic spline interpolation be able to handle responses from *stiff* systems?
>
> We thank the reviewer for bring up an interesting discussion point. To address the reviewer’s comment, we created simulation data from the Robertson’s chemical reaction system and constructed a cubic spline interpolant from it. The results are displayed in Figure 2 of the pdf linked in the global comment. The results show that in this case, cubic spline interpolation does return satisfying results, except for a slight ripple after the first peak in the y variable.
>
> Still, we do agree that it is possible that on some time series datasets, such as those from more pathological stiff systems or data with large amounts of noise, cubic spline interpolation may return sub-par results. However, our experiences seem to indicate that our training method is relatively robust to the quality of the interpolant.
>
> To support this point, we would like to guide the reviewer to Figure 19 and Figure 20 of our appendix (please look at the appendix.pdf file in the zip file of our Supplementary materials as it is the updated version compared to the one at the end of the main paper pdf), where we plot trained NeuralODE predictions, as well as the cubic spline interpolants used in the training for increasing noise in the data. We find that when the noise amplitude becomes large, the cubic spline interpolant displays high frequency artifacts (Figure 20, rightmost panel). However, the corresponding model predictions (Figure 19, rightmost panel) show no signs of this artifact carrying over, and still outperforms all other baseline methods.
>
> > I question the convergence of the homotopy method when constant $\Delta \lambda$ is used. The introduction of homotopy to the original ODE effectively increases the order of the ODE by 1, and transform it into a DAE. The authors should address whether there is the potential of non-convergence, and when it happens, how to address it.
>
> We thank the reviewer for bringing up another important point. We agree with the reviewer that our homotopy training method does not guarantee convergence to the global minimum. The reason for this is because homotopy optimization method, while effective, is still a local optimization method. One possible resolution to this problem is to use the HOPE algorithm from [2], which is a combination of homotopy optimization and particle swarm optimization. By employing multiple "particles" to explore the parameter space, the algorithm has a much higher chance to converge to the global optimum at the cost of increased computation.
>
> Finally, regarding the reviewer’s comment about the homotopy turning the original ODE into a DAE of 1 order higher, we are less familiar with DAEs and are unsure if this is indeed the case. The reason behind our confusion is that in the original ODE, the states evolve with time t, whereas the homotopy parameter changes with the training epochs. Therefore, we are not sure if our situation can be cast into the typical form of DAEs, given by
>
> $$
> \frac{d\bf u}{dt}={\bf f}(t, \bf u, v);\quad 0={\bf g}(t, \bf u, v)
> $$
>
>
> We would love to discuss more about this point and learn from the reviewer.
>
> [1] H. H. Robertson, The solution of a set of reaction rate equations, *Numerical Analysis: An Introduction*, pp. 178–182, Academic Press, London (1966).
>
> [2] D. M. Dunlavy and D. P. O'Leary, *Sandia Reports* SAND2005-7495 (2005).

---

> > ### Comment · Reviewer_jYQA · 2023-08-15
> > **Cubic Spline and others**
> >
> > Thanks to the authors for the responses and extra experiments.
> >
> > - Stiffness
> >
> > I am glad the authors are exploring the Robertson equation in the scipy package. However in my view it's not the extreme case. A good example can be found in Matlab discussion of stiff ODE's, such as stiff van der Pol equation with $\mu=1000$. I was wondering if cubic spline can handle the extremely sharp transition around $t=750$. (search for *matlab*, *solve-stiff-odes* to find the webpage).
> >
> > - Convergence
> >
> > I understand the authors' argument. I will come back to this issue again before the discussion period ends.

---

> > > ### Author Response · Authors · 2023-08-16
> > > **Additional results on the van der Pol equation**
> > >
> > > We thank the reviewer for kindly pointing us to the van der Pol equation, as well as the relevant references. We have replicated the code in the matlab documentation in Python and have used our cubic spline code to generate the interpolants. We present the results in the following anonymous link: https://drive.google.com/file/d/1xOZ3UnpUKmkB_X7ICuB226pgAprfTHK4/view?usp=sharing
> > >
> > > The results show that the cubic spline does incur some Gibb's phenomenon-like high frequency artifact at the waveform edges, but the severity of this artifact is not that large as witnessed by the zoomed-in figure in the right panel of our attached figure in the link above.
> > >
> > > We would also like to mention our noise studies on the Lotka-Volterra system  (Figure 19, 20 of our appendix: please look at the appendix.pdf file in the zip file of our Supplementary materials instead of the one at the end of the main paper pdf) where we found that oscillatory artifacts in the cubic spline does not carry over to the trained model predictions.
> > >
> > > We are happy to continue the discussion with the reviewer.

---

> > > > ### Comment · Reviewer_jYQA · 2023-08-20
> > > >
> > > > Thanks to the authors for the additional experiments. My view is that the cubic spline part can be further improved, but not necessarily a showstopper of the method. I believe the method is sound and there is sufficient value to the neurIPS audience.

---

### Official Review · Reviewer_9EGS · 2023-07-09

**Soundness:** 3 good
**Presentation:** 4 excellent
**Contribution:** 3 good
**Rating:** 6
**Confidence:** 3

**Summary:**

This paper presents a new training method for Neural Ordinary Differential Equations (NeuralODEs) that aims to improve their performance in extracting dynamical laws from time series data. The proposed method is based on synchronization and homotopy optimization, which does not require changes to the model architecture. The authors demonstrate that their method achieves competitive or better training loss while often requiring less than half the number of training epochs compared to other techniques. Furthermore, models trained with their method display better extrapolation capabilities.

**Strengths:**

[+] The paper is well-organized and clearly written, which makes it easy to follow and understand the proposed method. The introduction to the relevant background is sufficient, and the presentation of the experimental part is well-structured.

[+] The proposed method is well motivated, and some visualizations play a significant role in understanding the problems with traditional methods and the performance of the newly proposed method.

[+] The newly proposed method is very easy to integrate with models. In fact, the new method does not change the model but only changes the training process. It is simple to use and has excellent results.

[-] Considering that there are five main hyperparameters, if there are targeted sensitivity experiments in the experiments to demonstrate how the selection of hyperparameters affects the experimental results to some extent, the experimental results might be more convincing.

[-] The few systems involved in the experimental part are low-dimensional ODE systems. Can the proposed algorithm be applied to higher-dimensional problems? What new challenges might be encountered?

[?] What is the relationship between the K in the algorithm framework and the hyperparameters: Coupling strength (k)?

[?] It would be better if there is an analysis of computational complexity or a comparison of computation time with the baseline algorithm.


**Weaknesses:**

see above

**Questions:**

see above

**Limitations:**

see above

---

> ### Author Rebuttal · Authors · 2023-08-10
>
> We thank the reviewer for the detailed comments and suggestions. We have written our responses to some of the common questions in the global comment above, and have listed responses to the rest of the reviewer's questions below.
>
> > Considering that there are five main hyperparameters, if there are targeted sensitivity experiments in the experiments to demonstrate how the selection of hyperparameters affects the experimental results to some extent, the experimental results might be more convincing.
>
> While we agree with the reviewer, due to limited computational resources, we were only able to sweep a subset of the full hyperparameter space while running our experiments. Still, to provide some more information, we have included training curves for our preliminary runs where we individually varied some of the hyperparameters in Figure 3 of the pdf attached in the global comments. In case of the homotopy parameter decrement ration $\kappa$ and the number of homotopy steps, a suboptimal choice causes the train loss to jump in the later stages of the optimization due to the loss landscape changing too drastically after a homotopy step. In case of the learning rate, larger values cause the train loss to decrease rapidly in the initial stages, but also likely to stick in a bad local minimum later on during training. For the coupling strength, larger values cause the training loss to start from a smaller value and facilitate stable optimization. However, using too large values cause the coupled train loss to act completely independent of the true mean squared error due to insensitivity of the model parameters to the train loss.
>
> > It would be better if there is an analysis of computational complexity or a comparison of computation time with the baseline algorithm.
>
> First regarding the computational complexity, our homotopy algorithm has almost identical complexity as the vanilla training of NeuralODEs. This is because our algorithm only changes the differential equation to be solved (by adding on a coupling term to the model). The major factors of computational complexity are how the differential equation is solved and how the gradients are calculated (direct backpropagation or adjoint method-based techniques) - which were kept identical for both the homotopy and vanilla methods in our experiments.
>
> One difference between homotopy and vanilla training arises from the calculation of the cubic spline interpolant. However, this is calculated only once prior to the training and the resulting interpolations are cached for use during training. Therefore this only adds a negligible constant factor to the total algorithm.
>
> While this was not a direction explored in our paper, in the case of the multiple shooting, the computational complexity can be made different if one follows the approach of [1]. In this implementation, all of the segments for multiple shooting are solved in parallel, bringing down the computational complexity by a factor equal to the number of multiple shooting segments used. However, we would like to clarify that this does not always guarantee shorter training time there is a competition between each epoch of the multiple shooting method taking shorter time and our homotopy method requiring much less total epochs to converge. The use of adaptive solvers (which we describe below) further obfuscates this problem, and a further investigation is required for a final verdict.
>
> Regarding the computation time, we want to preface this point by saying that we did not account for such analysis at the time of our experiments, and thus our implementations are unfortunately not streamlined for speed. In the case of the homotopy method, some part of the cubic spline coefficients were recomputed each epoch and our multiple-shooting implementation was not parallelized in time. Furthermore, we had used multiple machines to run some of the longer experiments in parallel, which made it difficult to provide a comprehensive computational time benchmark. We apologize for this inconvenience and will improve on this point in future works.
>
> With this, we present the computation time benchmarks for the Lotka-Volterra training results in Table 1 of the attached pdf in the global comment. We see that while our homotopy method does take longer time per epoch than the other methods, the time reached to reach best epoch is much shorter due to the much less number of epochs required to train the model. A point worth stressing is that since we used 5 segments for the multiple shooting training, the expected speed up from parallelizing the multiple shooting algorithm is about 5x. However, our homotopy method takes less than this time to converge, indicating that the our method is competitive with even a optimized,  parallelized version of the multiple-shooting algorithm.
>
> Finally, we want to comment on one factor, other than the algorithm time complexity, that can greatly affect the computation time. This is due to the use of the adaptive ODE solver for solving the NeuralODE. As adaptive solvers evaluate the ODE variable amount of times in order to reach given tolerance values, a method that has a better per-epoch time complexity can show worse computation times if the solver evaluates the ODE more times for that method. This number of function evaluations is itself dependent on the form of the differential equation - i.e. the parameters of the NeuralODE. Therefore, a training method that guides the NeuralODE through a more favorable region of the parameter space will result in lesser number of function evaluations throughout training, which can result in reduced total training time even if the method’s per-epoch time complexity is worse than its competitors.
>
> [1] M. Stefano et. al, Differentiable Multiple Shooting Layers. NeurIPS 34 (2021).

---

> > ### Comment · Reviewer_9EGS · 2023-08-16
> >
> > Thanks for the author's response. I will keep my positive socre.

---

### Author Rebuttal · Authors · 2023-08-10

1. Can our algorithm be applied to potentially partially observed high dimensional data?

We believe that our algorithm can be applied to higher-dimensional problems, provided that a couple of challenges due to the increased dimensionality are addressed. Depending on the nature of the data, we believe there are two possible strategies.

The first strategy is to use a latent NeuralODE model where a NeuralODE is used to model the latent state dynamics ${\bf z}(t)$ via:
$\frac{d\bf z}{dt}={\bf U_1}(t, {\bf z};\bf\theta_1)$ with the latent state mapped to the observables ${\bf u}(t)$ by ${\bf u}(t)={\bf U_2}({\bf z}(t); \bf\theta_2)$. Depending on the available prior knowledge, ${\bf U_2}$ can either be a fixed equation, a parametrized equation, or a neural network.

Coupling between the dynamics and the data can achieved by the following augmentation:
$\frac{d\bf z}{dt}={\bf U_1}(t, {\bf z};{\bf\theta_1})-{\bf K}({\bf u}-{\bf \hat u})$ where $\bf \hat u$ is the interpolated version of the observed data. Such type of coupling has been reported in the ODE synchronization literature, an example being [1]. We expect this scheme to perform in the NeuralODE setting as well, provided that the observable $\bf u$ contains sufficient information about the latent state $\bf z$.

The second strategy to tackle observations from high-dimensional systems is to perform a phase space reconstruction using the observable time series, then train a simple NeuralODE on the reconstructed phase space data. In this case, our training method remains the same since we have access to all of the states of the reconstructed phase space. This idea was explored in [2], where the authors used temperature time series data from a weather station in Jena, Germany to train recurrent neural networks. As temperature corresponds to a scalar observable from an unknown high dimensional dynamical system, the data was detrended, denoised, then subjected to phase space reconstruction using time delay embedding.


2. Training on real world data

We thank the reviewer for the suggestion. To address the reviewer’s concerns and to continue the discussion in our previous response, we trained a NeuralODE on a sub-sampled version of this phase space reconstructed temperature dataset (to reduce computational burden) using our homotopy-based method. We present the results in Figure 1 of the attached pdf.

We find that our model can predict the reconstructed dynamics well in the interpolation interval, but fails in the extrapolation regime. This is likely due to insufficient portion of the high dimensional phase space being sampled by the training data, and should be improved with additional training data.

We find that our model can predict the reconstructed dynamics well in the interpolation interval, but fails in the extrapolation regime. This is likely due to insufficient portion of the high dimensional phase space being sampled by the training data, and should be improved with additional training data. Still, this result shows the feasibility of using our method on real world data, and we plan to continue exploring in this area.

3. Relationship between K and k
We apologize for the confusion. In our work, the matrix $\bf K$ and the scalar algorithm hyperparameter $k$ are related to each other by ${\bf K}=k\bf I$, where $\bf I$ is the identity matrix.

4.

---

### Decision · Program_Chairs · 2023-09-21

**Decision:**

Accept (poster)

**Comment:**

This is a solid submission. Novelty and relevance to the NeurIPS community have been acknowledged by all reviewers. There were multiple suggestions regarding he experimental validation of the method and we strongly encourage the authors to incorporate the feedback and especially the promised improvements in the final version of the paper.